# Neuronal variability and tuning are balanced to optimize naturalistic self-motion coding in primate vestibular pathways

Diana E Mitchell[1], Annie Kwan[1], Jerome Carriot[1], Maurice J Chacron[1], Kathleen E Cullen[1,2†*]

[1]Department of Physiology, McGill University, Montreal, Canada; [2]Department of Biomedical Engineering, The Johns Hopkins University, Baltimore, United States

**Abstract** It is commonly assumed that the brain's neural coding strategies are adapted to the statistics of natural stimuli. Specifically, to maximize information transmission, a sensory neuron's tuning function should effectively oppose the decaying stimulus spectral power, such that the neural response is temporally decorrelated (i.e. 'whitened'). However, theory predicts that the structure of neuronal variability also plays an essential role in determining how coding is optimized. Here, we provide experimental evidence supporting this view by recording from neurons in early vestibular pathways during naturalistic self-motion. We found that central vestibular neurons displayed temporally whitened responses that could not be explained by their tuning alone. Rather, computational modeling and analysis revealed that neuronal variability and tuning were matched to effectively complement natural stimulus statistics, thereby achieving temporal decorrelation and optimizing information transmission. Taken together, our findings reveal a novel strategy by which neural variability contributes to optimized processing of naturalistic stimuli.
DOI: https://doi.org/10.7554/eLife.43019.001

*For correspondence:
kathleen.cullen@jhu.edu

Present address: [†]Department of Biomedical Engineering, The Johns Hopkins University, Baltimore, Maryland, United States

Competing interests: The authors declare that no competing interests exist.

## Introduction

A fundamental challenge in neuroscience is to understand how the brain processes sensory information in order to generate accurate perception and guide appropriate behavioral responses. During everyday life, the inputs to our sensory systems have clearly defined statistical structures characterized by a pronounced decay in spectral power with increasing frequency (visual and auditory systems as reviewed in: *Simoncelli and Olshausen, 2001*; vestibular system: *Carriot et al., 2014*). It is generally agreed that sensory pathways are adapted to the statistical properties of natural stimuli (*Laughlin, 1981*; *Attneave, 1954*; *Barlow, 1961*; *Simoncelli and Olshausen, 2001*; *Olshausen and Field, 2004*; *Wark et al., 2007*). In this context, the brain is thought to optimize information transmission by removing the redundancy of (i.e. decorrelate) time-varying input, such that the spectral power of the resulting neuronal response is constant (i.e. are 'whitened'; *Srinivasan et al., 1982*; *Atick and Redlich, 1990*; *Atick, 2011*). Indeed, the fact that whitening has been reported in several early sensory pathways supports this idea (*Dan et al., 1996*; *Huang et al., 2016*).

The prevailing view is that whitening at the single neuron level is achieved as a result of a precise match between tuning and stimulus statistics. Specifically, it has been proposed that increases in the neuronal tuning function effectively compensate for the decaying spectral power of natural stimuli (*Dan et al., 1996*; *Wang et al., 2003*; *Pozzorini et al., 2013*; *Huang et al., 2016*). Such high-pass tuning is thought to be mediated by intrinsic cellular dynamics that give rise to spike frequency adaptation (*Benda and Herz, 2003*; *Wang et al., 2003*; *Pozzorini et al., 2013*) and/or other

mechanisms such as synaptic plasticity (*Jackman and Regehr, 2017*). However, in general, a match between the frequency spectra of neural tuning and stimulus statistics alone will not optimize information transmission. This is because the structure of the trial-to-trial variability of the neural response can strongly influence optimal coding as predicted from theory (*van Hateren, 1992*; *Rieke et al., 1996*; *Tkacik et al., 2008*; *Tkacik et al., 2010*). Further, in most brain areas, neural variability is not independent of frequency as it displays significant temporal correlations (*Jaeger and Bower, 1994*; *Gershon et al., 1998*; *Manwani and Koch, 1999*; *Oram et al., 1999*; *Goldberg, 2000*; *Maimon and Assad, 2009*; *Shinomoto et al., 2009*; *Massot et al., 2011*). To date, however, the effects of frequency-dependent structure of trial-to-trial variability on optimal coding at the single neuron level have for the most part been ignored (*Field, 1987*; *Körding et al., 2004*; *Graham et al., 2006*; *Huang et al., 2016*), not systematically investigated (*van Hateren et al., 2002*), or found to have minimal effect (*Pitkow and Meister, 2012*). Thus, the fundamental question of whether the frequency spectrum of neuronal variability actively contributes to optimizing information transmission of natural stimuli by sensory neurons has not yet been addressed experimentally.

The vestibular system benefits from well-described organization and easily described natural stimuli (e.g. head motion as a function of time; *Cullen, 2011*). This essential system generates reflexes that are vital for gaze and posture stabilization, as well as for accurate spatial perception and motor control (*Cullen, 2012*). Vestibular afferents displaying a wide range of resting discharge variabilities innervate the receptor cells of the vestibular sensors and synapse onto neurons within the central vestibular nuclei. These project to both motor centers and higher brain areas (e.g. thalamus; *Goldberg, 2000*; *Cullen, 2012*), thereby mediating vital reflexes and self-motion perception, respectively. Previous studies have emphasized the role of the resting discharge (i.e. in the absence of stimulation) on sensory processing across systems (for review, see: *Ringach, 2009*). Indeed, the variability of the resting discharge contributes to determining response variability during stimulation (*Ratnam and Nelson, 2000*; *Chacron et al., 2003*; *Chacron et al., 2005*; *Sadeghi et al., 2007a*). In particular, differences in resting discharge variability for afferents and central vestibular neurons strongly influence information transmission about self-motion, as assessed by using artificial (e.g. sinusoidal or noise) stimuli (*Sadeghi et al., 2007a*; *Massot et al., 2011*). However, how resting discharge variability contributes to central vestibular neural responses to natural self-motion remains an open question.

Here, we recorded the responses of both individual afferents and their post-synaptic central neuron targets within the vestibular nuclei to naturalistic self-motion stimuli. We found that early vestibular pathways demonstrate whitening at the first stage of central processing. Specifically, the response power spectra of central vestibular neurons are independent of frequency. Importantly, whitening is not simply inherited from afferents. Additionally, the tuning properties of central neurons alone do not predict their exceptionally whitened responses to natural self-motion. By using a computational model, we predicted that the frequency spectra of both tuning and variability will both significantly influence information transmission. Our experimental data confirmed modeling predictions of how neuronal variability affects information transmission. Notably, while increasing the level of variability will decrease information transmission, the frequency spectrum of neural variability for a given level will strongly determine optimality of coding (i.e. how close is the mutual information to its maximum value). Overall, we provide experimental evidence that the frequency spectrum of neural variability is matched with that of the tuning function in order to optimize coding during natural stimulation. Indeed, we found that coding was more optimized in central neurons than in afferents as the mutual information normalized by its maximum value was significantly higher. We hypothesize that these findings generalize across systems and species.

## Results

We recorded from central vestibular-only (VO) neurons within the vestibular nuclei (VN) that are responsive to rotational head motion but not eye movements (n = 27). These neurons receive direct synaptic input from vestibular afferents, and project to higher brain areas mediating self-motion perception (*Figure 1A*; reviewed in: *Cullen, 2012*). We found that central neurons displayed large and highly variable resting discharge rates (51.1 ± 2.8 sp/s; CV*=0.44 ± 0.03) in the absence of stimulation, consistent with prior characterizations (e.g. *Massot et al., 2011*). For comparison, recordings were also made directly from individual semicircular canal afferents that were classified as either regular (n = 14; CV*=0.06 ± 0.005) or irregular (n = 12; CV*=0.33 ± 0.05) based on a previously

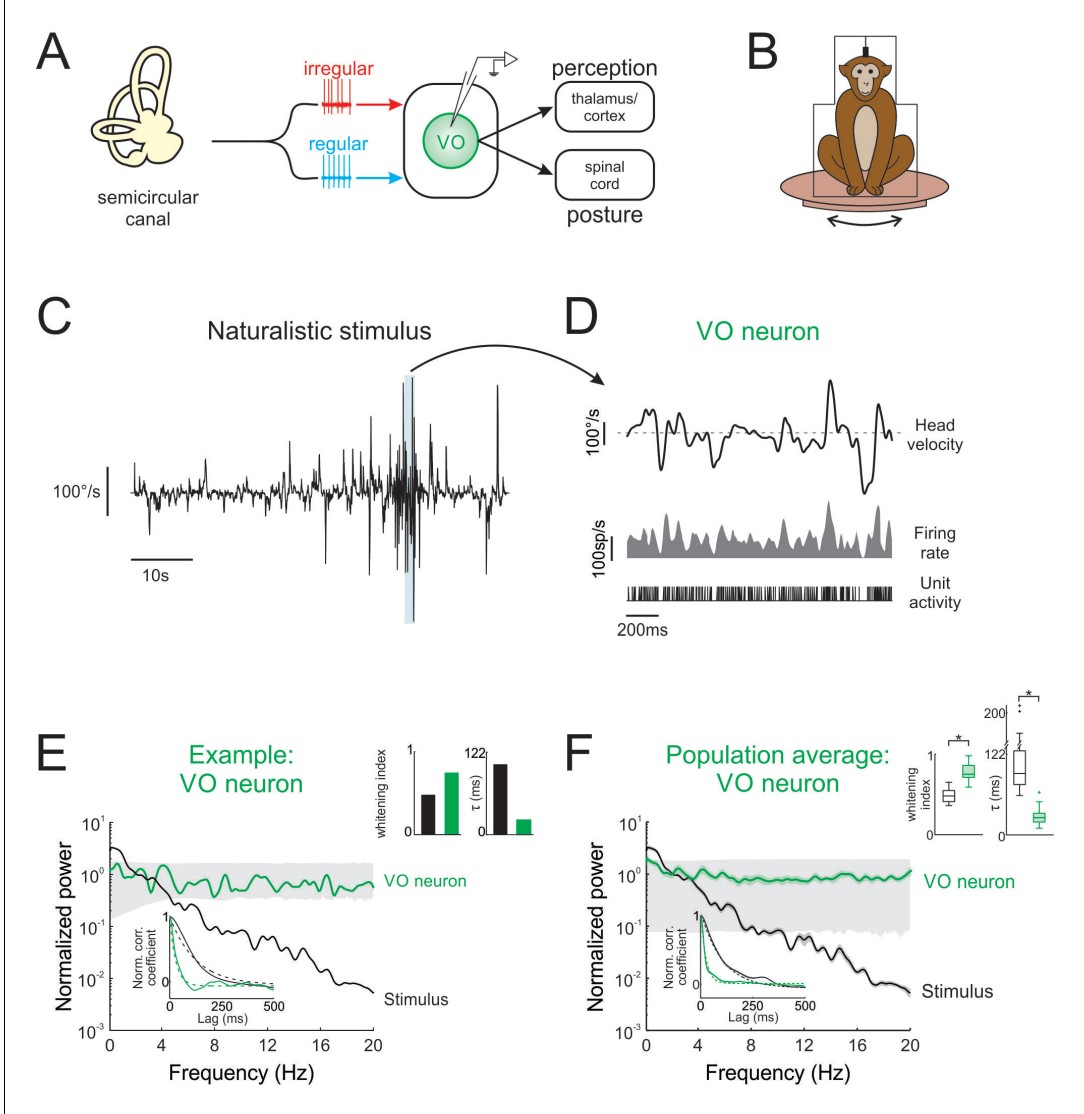

**Figure 1.** Central vestibular neurons optimally encode natural self-motion stimuli through whitening. (A) Vestibular afferents provide input to VO neurons in the vestibular nuclei, which in turn project to the spinal cord and higher order brain areas (i.e. thalamus/cortex). Recordings were made from VO neurons. (B) During experiments, the monkey was head-fixed and comfortably seated in a chair placed on a turntable. (C) Naturalistic head velocity during stimulation. (D) Response of an example VO neuron to the head velocity stimulus shown in C. (E) Normalized power spectra of natural stimuli and neural responses for an example VO neuron. Lower left insets shows autocorrelation functions (solid lines) and exponential fits (dashed lines) used to calculate correlation times (lower left), as well as correlation times and whitening indices (upper right). (F) Normalized power spectra of natural stimuli and population averages for VO neurons. Insets show autocorrelation functions (lower left), as well as correlation times and whitening indices (upper right). Whitening indices from VO neurons were significantly higher than those computed from the stimulus (Student's *t*-test, p < 0.001, t(26) = 10.9), while their correlation times were significantly lower (Student's *t*-test, p < 0.001, t(26) = 16.7). Gray bands show 95% confidence interval obtained using Poisson processes with the same firing rate as the experimental data for which the power spectrum is independent of frequency by definition (see Materials and methods). Error bars show ±1 SEM.

DOI: https://doi.org/10.7554/eLife.43019.002

The following figure supplement is available for figure 1:

**Figure supplement 1.** The self-motion stimuli used to elicit responses from vestibular neurons in our study closely matched head movement recordings from freely moving monkeys performing natural behaviors.

DOI: https://doi.org/10.7554/eLife.43019.003

established bimodal distribution of resting discharge variability corresponding to differences in axon diameter, response dynamics, and response sensitivity (*Goldberg, 2000*; *Sadeghi et al., 2007a*; *Eatock et al., 2008*; *Massot et al., 2011*; *Massot et al., 2012*). Notably, regular and irregular afferents displayed high resting discharge rates (regular: 102.4 ± 6.7 sp/s; irregular: 93.7 ± 8.9 sp/s), consistent with previous studies (e.g. *Sadeghi et al., 2007a*).

## Central vestibular neurons display temporal whitening in response to naturalistic self-motion

To test whether neurons within early vestibular pathways optimally encode natural self-motion statistics, we recorded their responses during naturalistic vestibular stimulation (*Figure 1B*) whose time-course closely mimicked that of natural self-motion signals (see Materials and methods and *Figure 1—figure supplement 1A*). Because the power spectra of these signals decayed by more than two orders of magnitude over the frequency range 0–20 Hz (*Figure 1—figure supplement 1A*), we restricted our analysis to this frequency range. We found that central vestibular neurons showed robust responses to naturalistic self-motion stimuli (*Figure 1D*, bottom panels).

If central vestibular neurons optimally encode natural stimuli, then their response power spectra should be constant as a function of temporal frequency (i.e. white). The response power of an example central vestibular neuron is shown together with that of the stimulus in *Figure 1E* (green and black curves, respectively). Indeed, the response power spectrum was constant for frequencies within the physiologically relevant range (0–20 Hz; *Huterer and Cullen, 2002*; *Carriot et al., 2017*). In contrast, the stimulus power spectrum strongly decayed with increasing frequency (*Figure 1E*, black curve). Results were quantified by computing a whitening index (see Materials and methods) that is maximum and equal to unity for a constant spectrum. We found that this example neuron displayed a higher whitening index than the stimulus (*Figure 1E*, left panel of top right inset). We also evaluated to what extent whitening in the frequency domain corresponded to decorrelation in the temporal domain. Specifically, if whitening occurs in the frequency domain (i.e. the power spectrum of the response is more independent of frequency than that of the stimulus), then the width of the response autocorrelation function must decrease. As expected, the response autocorrelation function correlation time (~28 ms) was much lower than that of the stimulus (~106 ms; *Figure 1E*, right panel of top right inset).

Overall, analysis of our entire central neuron data gave rise to qualitatively similar results: while the stimulus power spectrum decayed rapidly for frequencies greater than 2 Hz, the response power spectrum instead remained constant as a function of frequency (*Figure 1F*, compare black and green curves, respectively). The response whitening index values were significantly higher than those obtained from the stimulus (*Figure 1F*, left panel of top right inset; p < 0.001). Correspondingly, in the temporal domain, the response autocorrelation function decayed to zero faster than that of the stimulus (*Figure 1F*, bottom inset, compare green and black curves, respectively), as quantified by significantly lower correlations times (*Figure 1F*, right panel of top right inset; p < 0.001). Thus, taken together, our results show that central vestibular neurons optimally encode natural self-motion stimuli through whitening.

## Peripheral afferents are not sufficient to account for whitening by central vestibular neurons

The simplest explanation for our results shown above is that afferent responses to natural self-motion stimuli are already temporally whitened and that this is simply transmitted to central vestibular neurons. To test this, we next recorded afferent responses to naturalistic self-motion (*Figure 2A*). We considered the responses of both regular and irregular afferents that each innervate the canal endorgans and serve as parallel channels that project to central vestibular neurons (*Highstein et al., 1987*; *Boyle et al., 1992*). We found that, while regular and irregular afferents showed robust responses to naturalistic self-motion stimuli (*Figure 2B*, left and right panels, respectively), neither optimally encode naturalistic self-motion stimuli as their power spectra were not independent of frequency (*Figure 2C,D,E,F*). First, *Figure 2C* shows the stimulus (black curve) and response (blue curve) power spectra for a typical regular afferent. The response power spectrum decayed as a function of frequency, largely overlapping that of the stimulus (*Figure 2C*). Correspondingly, in the temporal domain, there was also strong overlap between the response and stimulus autocorrelation

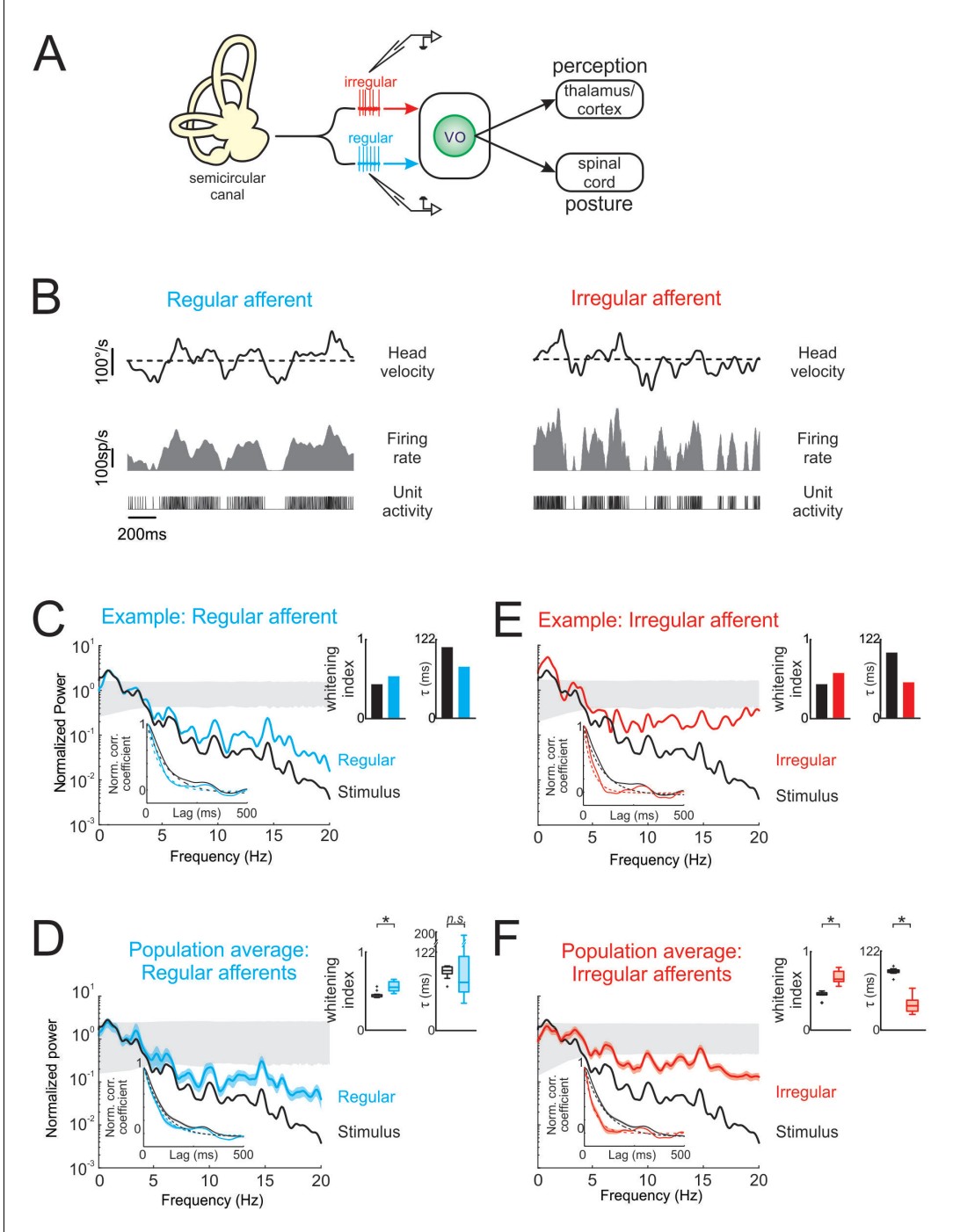

**Figure 2.** Vestibular afferents do not perform whitening of natural self-motion. (A) Same diagram as in *Figure 1A*, except that recordings were made from afferents. (B) Response of example regular (left) and irregular (right) afferents during natural stimulation. (C) Normalized power spectra of natural stimuli (black) and neural response (blue) for the example regular afferent. The lower left inset shows the autocorrelation function of the response (blue) and of the stimulus (black). Solid lines represent autocorrelation functions, while the dashed lines are exponentials fits. The upper right inset shows the whitening index (left) and correlation time (right). (D) Normalized population-averaged power spectrum for regular afferents (blue) with that of the stimulus (black). The lower left inset shows the population-averaged response autocorrelation function (blue). The upper right inset shows the population-averaged whitening index (left) and correlation time (right) values. The whitening index was not significantly different than that of the stimulus (Student's $t$-test, p = 0.79, t(13)=0.27). The correlation time was significantly lower than that of the stimulus (Student's $t$-test, p < 0.001, t(13) =2.9). (E) Normalized power spectra of natural stimuli (black) and neural response (red) for the example irregular afferent. The lower left inset shows the autocorrelation function of the response (red) and of the stimulus (black). Solid lines represent autocorrelation functions, while the dashed lines are exponentials fits. The upper right inset shows the whitening index (left) and correlations time (right). (F) Normalized population-averaged power

*Figure 2 continued on next page*

*Figure 2 continued*

spectrum for irregular afferents (red) with that of the stimulus (black). The lower left inset shows the population-averaged response autocorrelation function (red). The upper right inset shows the population-averaged whitening index (left) and correlation time (right) values. The whitening index was significantly higher than that of the stimulus (Student's *t*-test, p < 0.001, t(11)=2.25). The correlation time was significantly lower than that of the stimulus (Student's *t*-test, p < 0.001, t(11)=9.64). Gray bands show 95% confidence interval obtained using Poisson processes with the same firing rate as the experimental data for which the power spectrum is independent of frequency by definition (see Materials and methods). Error bars show ±1 SEM.

DOI: https://doi.org/10.7554/eLife.43019.004

functions (*Figure 2C*, lower inset). Accordingly, both the whitening index and correlation time values computed from the response and the stimulus were comparable (*Figure 2C*, top inset). Overall, qualitatively similar results were seen across our regular afferent dataset (*Figure 2D*). The population-averaged response power spectrum and autocorrelation function also largely overlapped with those of the stimulus (*Figure 2D*, compare blue and black curves in the main panel and in the inset). The population-averaged response whitening index value was slightly larger than that of the stimulus (*Figure 2D*, left panel of top right inset; p < 0.001) but still much smaller than that computed for central vestibular neurons (compare with *Figure 1E*, left panel of top right inset). Further, the population-averaged response correlation time value was not significantly different than that of the stimulus (*Figure 2D*, right panel of top right inset; p = 0.79). Thus, our results indicate that, unlike central vestibular neurons, regular afferents do not perform whitening of natural self-motion stimuli.

Second, a comparable analysis of irregular afferent responses revealed qualitatively similar results. *Figure 2E* shows the stimulus (black curve) and response (blue curve) power spectra for a typical irregular afferent, while *Figure 2F* shows the population-averages. Overall, the response power spectra of irregular afferents were also not constant as a function of frequency (*Figure 2E and F*, red curves), although the power spectra of irregular afferents decayed more slowly than those of their regular counterparts (compare *Figure 2D and F*). Further, the response autocorrelation functions did not rapidly decay to zero (*Figure 2E and F*, red curves in lower left insets). Thus, our results indicate that, like regular afferents but unlike central vestibular neurons, irregular afferents did not perform whitening of natural self-motion stimuli, as their response power spectra were not constant as a function of frequency.

## Temporal whitening in early vestibular pathways occurs in stages

Thus far, our results have shown that the response power spectra of central vestibular neurons but not of afferents are constant as a function of frequency. Therefore, whitening of natural self-motion by central vestibular neurons is not simply inherited from their afferent input. To systematically compare whitening by regular afferents, irregular afferents, and central vestibular neurons, we superimposed their power spectra in *Figure 3A*. Comparison of the population-averaged whitening indices (*Figure 3B*) revealed that significantly higher values for central vestibular neurons than for either class of afferents (p < 0.01; *Figure 3B*). However, we found that whitening index values for irregular afferents were significantly higher than those for their regular counterparts (p = 0.002; *Figure 3B*). Correspondingly, to facilitate comparison in the time domain, the population-averaged autocorrelation functions of regular afferents, irregular afferents, and central vestibular neurons are superimposed (*Figure 3C*). Consistent with our spectral analysis above, population-averaged correlation times for central vestibular neurons were lower than for either class of afferents (regular: p < 0.001; irregular: p = 0.004; *Figure 3D*), while correlation times for irregular afferents were significantly lower than those of regular afferents (p = 0.006; *Figure 3D*). Thus, we conclude that whitening occurs in stages: some decorrelation of naturalistic self-motion input is achieved by irregular afferents and further decorrelation of this input occurs as the level of central vestibular neurons.

## The tuning properties of central vestibular neurons are not sufficient to explain whitening of naturalistic self-motion stimuli

As described above, the prevailing view is that sensory pathways achieve whitening by matching neural tuning to stimulus statistics. Specifically, increases in the neuronal tuning function should effectively compensate for the decaying stimulus power spectrum, such that the response power spectrum is constant as a function of frequency (*Figure 4A*). To test whether whitening by central

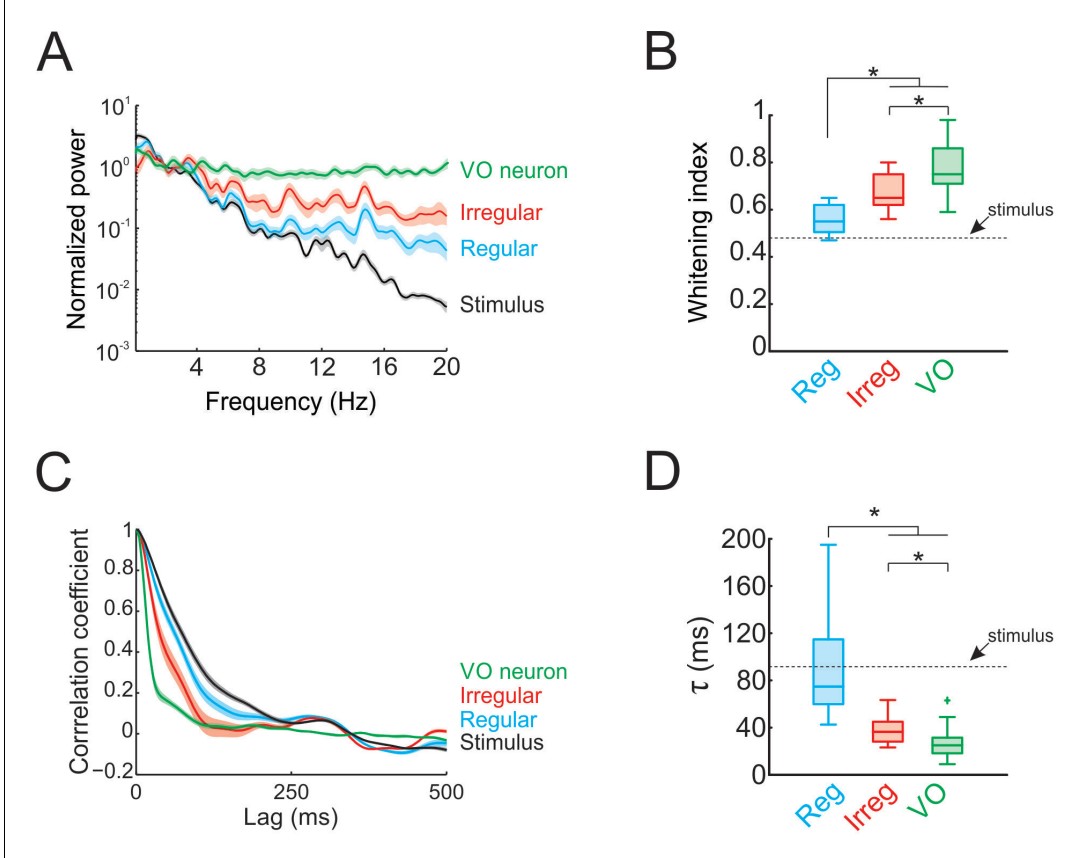

**Figure 3.** Temporal whitening of neuronal responses occurs sequentially throughout early vestibular pathways. (**A**) Normalized power spectra of natural stimuli (black), and neuronal responses of regular (blue), irregular afferents (red), and VO neurons (green). (**B**) Population-averaged whitening index values for regular afferents (blue, left), irregular afferents (red, middle), and VO neurons (green, right). Values for VO neurons were significantly higher than those of either regular or irregular afferents, while those of irregular afferents were significantly higher than those of regular afferents (one-way ANOVA, p < 0.001, F(2,50)=24.26). (**C**) Normalized autocorrelation functions of natural stimuli (black), and neuronal responses of regular (blue), irregular afferents (red), and VO neurons (green). (**D**) Population-averaged correlation time values for regular afferents (blue, left), irregular afferents (red, middle), and VO neurons (green, right). Values for VO neurons were significantly lower than those of either regular or irregular afferents, while those of irregular afferents were significantly lower than those of regular afferents (one-way ANOVA, p < 0.001, F(2,50)=26.68).

DOI: https://doi.org/10.7554/eLife.43019.005

vestibular neurons can be explained based on their tuning properties, we first characterized their sensitivity to naturalistic self-motion as a function of frequency (*Figure 4B*, inset). We then used these tuning curves (i.e. sensitivity as a function of frequency) to predict the response power spectrum to naturalistic self-motion (see *Figure 4A* and Materials and methods). Comparison of the population-averaged predicted and actual response power spectra revealed a poor match (*Figure 4B*, compare dashed green and solid green curves). Quantification of our results showed that predicted whitening index values were consistently lower than actual values (p < 0.001; *Figure 4C*). Further, whitening by central vestibular neurons could not be accounted for by static nonlinearities (*Figure 4—figure supplement 1* and *Figure 4—figure supplement 2*; see Materials and methods), or by their tuning to artificial sinusoidal stimulation at discrete frequencies (*Figure 5*, see Materials and methods). In contrast, the tuning functions of both regular and irregular afferents were sufficient to predict their response power spectra to naturalistic self-motion (*Figure 4—figure supplement 3*), thereby showing that greater temporal whitening of naturalistic self-motion stimuli by irregular afferents is due to their greater high-pass tuning properties. Thus, we conclude that the tuning function of central vestibular neurons does not fully compensate for the decaying power spectrum of natural self-motion stimuli and thus cannot account for their whitened responses.

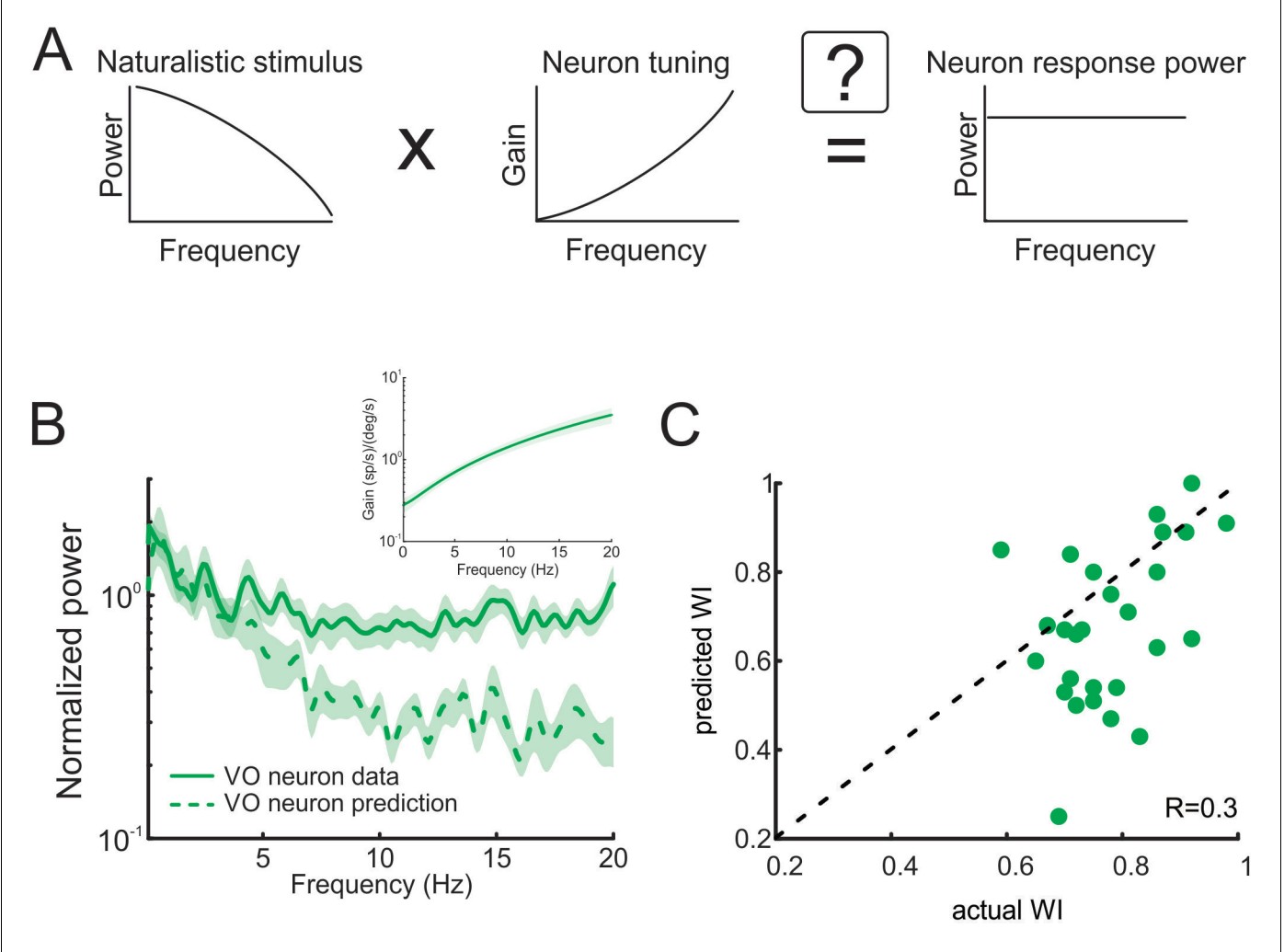

**Figure 4.** Neural tuning to naturalistic stimulation does not account for the temporally whitened responses of central vestibular neurons to natural stimuli. (**A**) Schematic showing how the response to a natural stimulus (right) is assumed to be determined from the stimulus spectrum (left) and the neural tuning function (middle). (**B**) Normalized population-averaged actual (solid green) and predicted (dashed green) response power spectra for central vestibular neurons. Inset: Population-averaged tuning curve showing gain as a function of frequency for central vestibular neurons. (**C**) Predicted versus actual whitening indices for central vestibular neurons. Most data points were below the identity line (dashed black).

DOI: https://doi.org/10.7554/eLife.43019.006

The following figure supplements are available for figure 4:

**Figure supplement 1.** Adding a static nonlinearity does not account for responses of central vestibular neurons to natural stimuli.
DOI: https://doi.org/10.7554/eLife.43019.007
**Figure supplement 2.** The residual between the actual firing rate and the nonlinear prediction does not account for responses of central vestibular neurons to natural stimuli.
DOI: https://doi.org/10.7554/eLife.43019.008
**Figure supplement 3.** Neural tuning to artificial sinusoidal stimulation and to natural stimulation accounts for responses of afferents to natural stimuli.
DOI: https://doi.org/10.7554/eLife.43019.009
**Figure supplement 4.** Mutual information is determined from both the noise and transmitted power spectra.
DOI: https://doi.org/10.7554/eLife.43019.010

## Effects of resting discharge variability and tuning on optimized information transmission through whitening

Theory predicts that the response power spectrum to natural stimuli is not solely determined by the tuning function (*Shannon, 1948*; *Rieke et al., 1996*). Specifically, the response power spectrum is equal to the sum of the noise power spectrum (i.e. the power spectrum of the neural variability) and

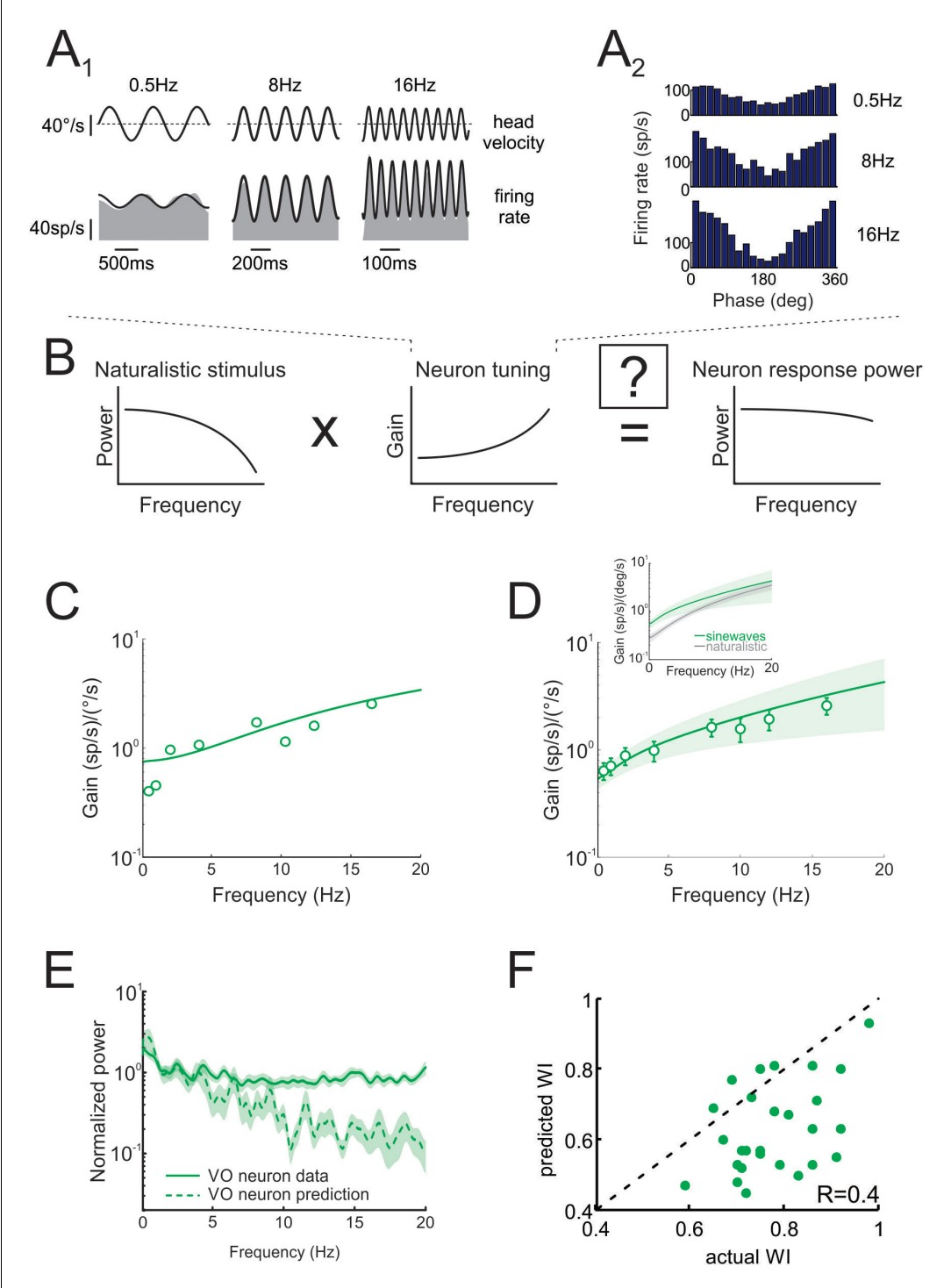

**Figure 5.** Neural tuning to artificial sinusoidal stimulation does not account for responses of central vestibular neurons to natural stimuli. (**A₁**) Response of example central vestibular neuron to sinusoidal yaw rotations with frequencies 0.5, 8 and 16 Hz. (**A₂**) Post-stimulus time histogram (PSTH) of this example central vestibular neuron to sinusoidal yaw rotations with frequencies 0.5, 8 and 16 Hz. (**B**) Schematic showing how the response to a natural stimulus (right) is assumed to be determined from the stimulus spectrum (left) and the neural tuning function (middle). (**C**) Gain as a function of frequency for an example central vestibular neuron with best fit (solid line). (**D**) Population-averaged gain as a function of frequency. The solid line is the best fit with bands showing ±1 SEM. (**E**) Normalized power spectra of neural responses (solid green) and predictions (dashed green) for central vestibular neurons. Inset: Population-averaged tuning curves obtained using naturalistic (gray) and sinusoidal (green) stimulation. Note that the tuning

*Figure 5 continued on next page*

*Figure 5 continued*

obtained using naturalistic stimulation was lower than that obtained using sinusoidal stimulation at low frequencies, consistent with the fact that central vestibular neurons display a boosting nonlinearity (*Massot et al., 2012*). (**F**) Predicted versus actual whitening indices.

DOI: https://doi.org/10.7554/eLife.43019.011

the power spectrum of the response that would be predicted by tuning alone (i.e. transmitted power; *Figure 4—figure supplement 4A*). Thus, in order to maximize information through whitening, it is the response power spectrum that should be constant as a function of frequency, rather than the transmitted power spectrum.

To better understand how the noise and the transmitted power spectra interact in order to influence information transmission, we simulated a simple model (see Materials and methods). We initially considered a given noise spectrum (*Figure 4—figure supplement 4B*, black curve) and systematically varied the rate at which the transmitted power spectrum increased while keeping the area under the curve constant (*Figure 4—figure supplement 4B*, green curves). When the increase in the transmitted power fully compensated for the decrease in noise power (*Figure 4—figure supplement 4B*, condition b, solid green), the response power was constant as a function of frequency (*Figure 4—figure supplement 4C*, condition b, solid light green) and the mutual information was maximum (*Figure 4—figure supplement 4D*, point b). In contrast, when the increase in the transmitted power either undercompensated (Fig. *Figure 4—figure supplement 4B*, condition a, thin dashed green) or overcompensated (Fig. *Figure 4—figure supplement 4B*, condition c, thick dashed green) the decrease in the noise power, the response power was not constant as a function of frequency and the mutual information was less than the maximum value (Fig. *Figure 4—figure supplement 4D*, points a and c, respectively). Thus, our results show that, for a given noise intensity and spectrum, varying the transmitted power spectrum can have significant influence on mutual information.

We then considered the situation for which the transmitted power spectrum is fixed and increased the noise intensity (*Figure 4—figure supplement 4E*). We found that, with increasing noise intensity, noise as well as response power increased uniformly for all frequencies (*Figure 4—figure supplement 4E,F*), but that mutual information decreased (*Figure 4—figure supplement 4G*). This is expected given that mutual information is related to the signal-to-noise ratio, which decreases with increasing noise intensity. *Figure 4—figure supplement 4H* illustrates the effects of systematically varying the rate at which the transmitted power increases as well as the noise intensity. Note that the conditions illustrated in *Figure 4—figure supplement 4B–D* are denoted as a horizontal white line, while those illustrated in *Figure 4—figure supplement 4E–G* are denoted as a vertical white line, respectively. Thus, taken together, our modeling results show how both the noise and the transmitted power spectra interact in order to affect information transmission. Specifically, to maximize information, the spectral frequency content of the neural variability should perfectly compensate the transmitted power, such that the sum of the two is constant as a function of frequency.

### Whitening of natural self-motion stimuli by central vestibular neurons can be explained by taking into account both their tuning properties and their resting discharge variability

We next tested our modeling predictions in the vestibular system. Theory predicts that the response power should then be equal to the sum of the noise power and of the transmitted power spectra (*Figure 6A*; *Risken, 1996*). Since central vestibular neurons display a substantial and highly variable resting discharge (e.g. *Massot et al., 2011*), we first used the power spectrum of the resting discharge in order to estimate that of trial-to-trial variability during stimulation. We found that the resting discharge power spectrum increases at higher frequencies (*Figure 6B*, inset, black curve) and thus, in conjunction with the neural tuning function (*Figure 4B*, inset), effectively compensated for decaying stimulus power. Indeed, the predicted and actual response power spectra were in excellent agreement (*Figure 6B*, compare green and light green curves). Further, predicted and actual whitening index values were not significantly different from one another (p = 0.39; *Figure 6C*). To confirm that the resting discharge is an accurate measurement of trial-to-trial variability during stimulation,

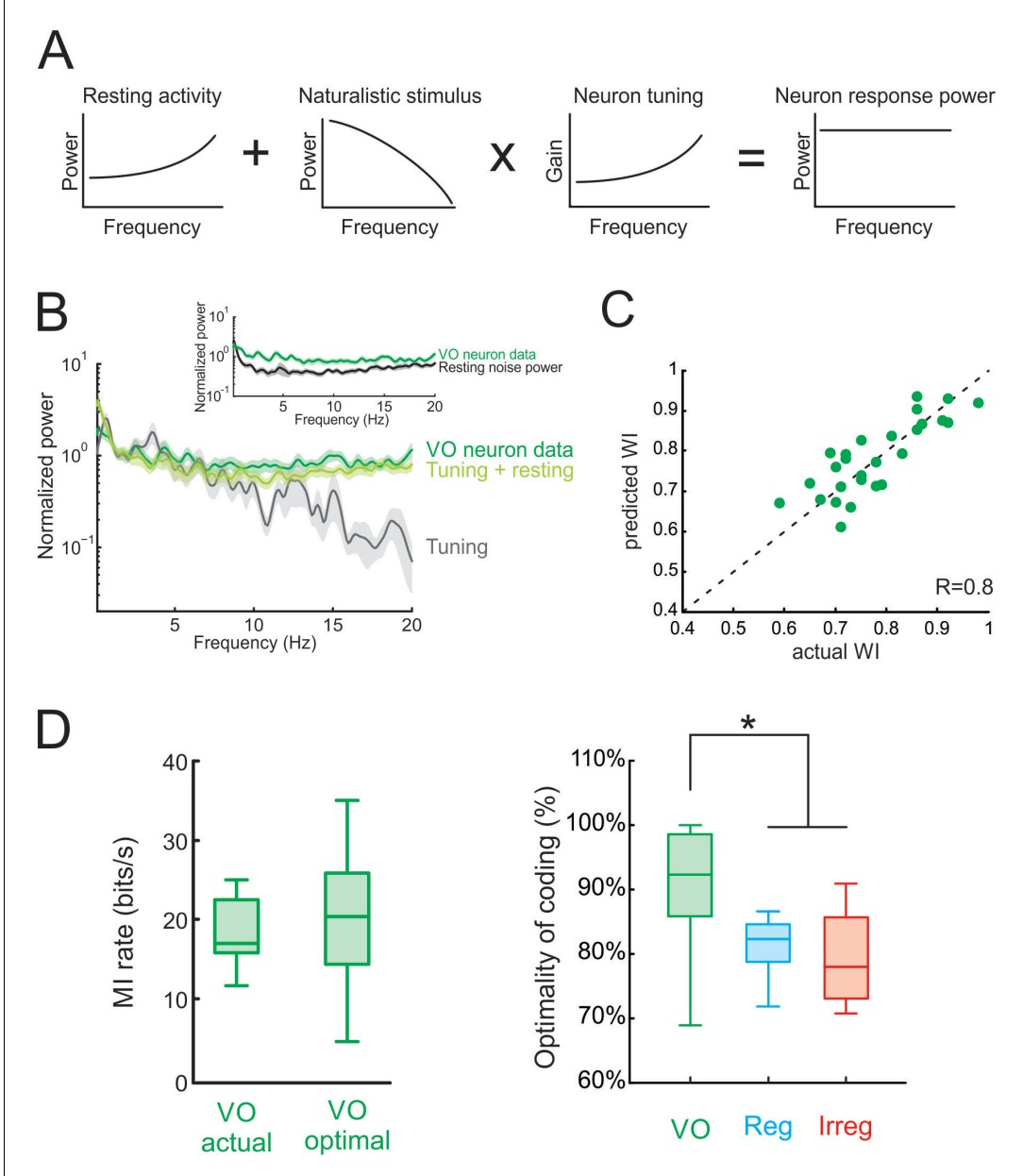

**Figure 6.** Neural variability and tuning determine whitening of natural self-motion stimuli by central vestibular neurons. **(A)** Schematic showing how the response power spectrum (right) can be predicted from the resting activity power spectrum (left) as well as the stimulus power spectrum (middle left) and the neural tuning function (middle right). Specifically, the resting activity power spectrum should compensate the transmitted power spectrum (i.e. that obtained by multiplying the stimulus power by the tuning function) such that the response power spectrum (i.e., the sum of the two) is constant as a function of frequency. **(B)** Population-averaged actual response power spectrum (dark green) together with the prediction from the resting discharge and tuning (light green). The transmitted power spectrum (grey) is also shown. The inset shows the population-averaged resting discharge (black) and actual response (dark green) power spectra. **(C)** Predicted versus actual whitening index values for central vestibular neurons. Data points were scattered across the identity line. **(D)** *Left*: Population-averaged actual (left) and maximum (right) information rate values for central vestibular neurons. *Right:* The population-averaged mutual information normalized by the optimal value for VO neurons (green) was significantly greater than that of regular (blue) and irregular (red) afferents (one-way ANOVA, p < 0.001, F(2,50)=14.79). Error bars represent ±1 SEM.

DOI: https://doi.org/10.7554/eLife.43019.012

The following figure supplements are available for figure 6:

**Figure supplement 1.** Predicting response power spectra of central vestibular neurons using trial-to-trial variability.

DOI: https://doi.org/10.7554/eLife.43019.013

*Figure 6 continued on next page*

*Figure 6 continued*

**Figure supplement 2.** Actual and maximum population-averaged mutual information rate values for central vestibular neurons (green) as well as regular (blue) and irregular (red) afferents.

DOI: https://doi.org/10.7554/eLife.43019.014

we quantified the trial-to-trial variability of the neural response to naturalistic self-motion for a subset of neurons (see Materials and methods). We found that the power spectrum of the resting discharge and the residual were quite similar (*Figure 6—figure supplement 1A*), thus validating our assumption that the power spectrum of the resting discharge can be used to estimate that of the trial-to-trial variability during stimulation. Consequently, using either the resting discharge or trial-to-trial variability accurately predicted the temporally whitened neural responses to naturalistic stimuli (*Figure 6—figure supplement 1B–C*). We further found that the response power spectra during naturalistic self-motion were well predicted by summing the power spectrum of the trial-to-trial variability and the trial-averaged response spectrum (*Figure 6—figure supplement 1*).

Finally, we quantified optimal coding by central vestibular neurons by comparing their rates of information transmission to optimal values (see Materials and methods). We found that actual information rates transmitted by central vestibular neurons were in good agreement with optimal values (*Figure 6D*, left panel), and comparison with afferents revealed that the actual information rate of central vestibular neurons was significantly closer to its maximum possible value (*Figure 6D*, right panel and *Figure 6—figure supplement 2*). Moreover, our modeling correctly predicted that, due to their higher variability, information rates computed from central vestibular neurons were lower than those of afferents (*Figure 6—figure supplement 2*). Thus, we conclude that the experimentally observed whitening by central vestibular neurons is achieved because both the power spectra of their variability and their tuning function effectively compensate for the decaying power of natural self-motion. Taken together, our findings show that the power spectrum of neural variability can have important consequences on determining whether sensory neurons optimally encode natural stimuli; thereby overturning the prevailing view that whitening is achieved by neural tuning alone.

## Discussion

### Summary of results

Here, we show that neurons at the first central stage of vestibular processing optimally encode natural self-motion stimuli through whitening due to a match between their frequency tuning and response variability. Specifically, we found that the spectral power of neuronal responses to naturalistic self-motion was constant as a function of temporal frequency. Given that we did not observe such whitening in their afferent input, we hypothesized that whitening occurs as a result of response properties, which are inherent to the central neurons. We thus first tested the prevailing view that whitening is a result of neural tuning, by establishing whether the frequency-dependent sensitivity of central neurons could account for the observed whitening. However, we found that the responses of vestibular neurons to naturalistic self-motion could not be explained based on their tuning alone. We hypothesized that a precise match between stimulus statistics, neural tuning, and response variability accounts for this unexpected result. To test this, we formulated a model that explicitly considered the dependence of both neuronal response variability and tuning on temporal frequency, and then demonstrated the generality of this model by predicting the spectra of neuronal responses to naturalistic stimulation. We then confirmed that this model could explain our neurophysiological findings, by showing that indeed response power predictions made using both resting activity and tuning were in excellent agreement with our experimental data. Taken together, these results both demonstrate that the responses of central vestibular neurons are optimal, and provide experimental evidence that, in early vestibular pathways, whitening (i.e. efficient coding) requires a precise match between the input distribution, neural tuning, and neural variability.

## Impact of resting discharge variability on coding of self-motion in the vestibular system

The vestibular system is well-suited for understanding how neural variability affects the coding of information. Theory predicts that the trial-to-trial variability during stimulation can be well approximated by that of the resting activity in the absence of stimulation (*Risken, 1996*) and our results show that this is the case for central vestibular neurons under naturalistic stimulation. Further, it has long been known that the resting rates of central vestibular neurons, despite being lower than those of afferents (~50 sp/s vs. ~100 sp/s, respectively), are substantially more variable (*Goldberg, 2000*; *Massot et al., 2011*). Previous theoretical studies have emphasized the role of the resting discharge on information transmission (*Chacron et al., 2004*; *Chacron et al., 2005*). Notably, sensory stimulation must perturb the resting discharge in order to be detected. In particular, the power spectrum of the resting discharge can be considered as a 'noise spectrum' as it represents the amount of noise present at each frequency, and thus can approximate neural variability under stimulation (*Ratnam and Nelson, 2000*; *Chacron et al., 2003*; *Chacron et al., 2005*; *Massot et al., 2011*; *Jamali et al., 2013*). In the vestibular system, the fact that central vestibular neurons display higher levels of resting discharge than afferents thus provides an explanation as to why they display higher detection thresholds, and transmit less information about artificial self-motion stimuli than single afferents (*Massot et al., 2011*). Consequently, the increased resting discharge variability observed at higher stages of vestibular processing is detrimental to information transmission. This leads to the question of how central vestibular neurons encode self-motion stimuli in order to mediate self-motion perception during natural behaviors.

Our results obtained using naturalistic self-motion stimuli are consistent with prior studies showing that central vestibular neurons transmit less information than single afferents. Our modeling predicts that this is due to their greater levels of resting discharge variability. Importantly, our results show that the spectral content of neural variability strongly influences information transmission. Thus, in order to optimize neural coding of natural self-motion stimuli through whitening, such that response power is constant as a function of frequency, it is essential that the spectral frequency content of the resting discharge perfectly compensate the power transmitted from the stimulus (i.e. the transmitted power, *Figure 4—figure supplement 4*). This theoretical concept has been previously referred to as the 'water filling analogy' where the transmitted power is distributed to 'fill in' a vessel shaped as the variability power (*de Ruyter van Steveninck and Laughlin, 1996*; *Rieke et al., 1996*). Our results provide experimental evidence for this concept in the context of optimizing information transmission of natural sensory input. Thus, while the higher resting discharge variability of central vestibular neurons actually decreases information transmission relative to that of afferents (*Figure 6—figure supplement 2*), the spectral content of this variability is set such as to optimize the information being transmitted since the mutual information for central vestibular neurons is closer to its maximum value than for afferents (*Figure 6D*).

This leads to the question: what are the implications of increased neural variability in central vestibular pathways for naturalistic self-motion coding? To answer this question, it is important to consider that, to be useful to an organism, transmitted information must be decoded by downstream brain areas. We hypothesize that a match between stimulus statistics, neural tuning, and trial-to-trial variability further optimizes information transmission at the population level as proposed in other systems (*Doi et al., 2012*; *Kastner et al., 2015*). Further studies involving multi-unit recording from central vestibular neurons however will be needed to understand how optimized coding of natural self-motion by single central vestibular neurons observed here affects population coding as well as subsequent decoding of the population response by downstream neurons.

## Role of variability in the coding of natural stimuli in other systems

Neural variability is seen ubiquitously in the central nervous system (for review see: *Stein et al., 2005*). Previous studies performed in multiple systems have shown that neural variability actually increases at the level of central neurons relative to that observed in more peripheral areas (visual: *Kara et al., 2000*; auditory: *Wang et al., 2008*; electrosensory: *Maler, 2009*), which is thought to be due to intense synaptic bombardment (*Destexhe et al., 2001*; *Destexhe et al., 2003*). However, the functional role of variability in neural coding continues to be the focus of much debate (*Stein et al., 2005*; *McDonnell and Ward, 2011*). Here, we showed that there is a match between

the tuning properties and variability of central vestibular neurons which enables optimized encoding of natural self-motion through temporal whitening. This result constitutes an experimental demonstration that the spectral frequency content of neural variability is matched to that of the tuning function to optimize encoding of natural stimuli, as predicted by theoretical studies (*Rieke et al., 1996*). Indeed, most previous studies have focused on tuning and either assumed that neural variability was negligible or did not display temporal correlations (*Dan et al., 1996*; *Wang et al., 2003*; *Pitkow and Meister, 2012*). However, neurons displaying large and variable spontaneous activity (i.e. resting discharge activity) with temporal correlations are found ubiquitously within the sensory periphery and central brain regions (*Hubel, 1959*; *Evarts, 1964*; *Pfeiffer and Kiang, 1965*; *Steriade et al., 1978*; *Jaeger and Bower, 1994*; *Köppl, 1997*; *Aizenman and Linden, 1999*; *Schneidman et al., 2006*; *Greschner et al., 2011*). Accordingly, we speculate that our results showing that neural variability, together with tuning, plays a fundamental role towards optimizing information transmission about natural stimuli, will be generally applicable across sensory systems and species.

### Future directions

In this study, we considered the coding of naturalistic self-motion stimuli that were passively applied along the yaw rotation axis. However, it is important to note that, under natural conditions, vestibular stimulation results from both active and passive self-motion. Previous studies have established that afferents respond similarly to both classes of stimuli (*Cullen and Minor, 2002*; *Sadeghi et al., 2007b*; *Jamali et al., 2009*). Thus, the results presented here for afferents are expected to be applicable to conditions where self-motion is actively generated. However, the central vestibular neurons which were the focus of the present study (i.e. VO neurons) display markedly attenuated responses to active self-motion consisting of head and/or body orienting movements (reviewed in: *Cullen, 2012*) due to integration of vestibular and extra-vestibular (e.g., proprioceptive and motor) signals. Thus, further studies will be needed to establish how these encode natural active self-motion signals (e.g. those experienced during locomotion). We further note that natural self-motion stimuli are generally not restricted to one axis of motion, but rather comprise of both three-dimensional rotational and translational components (*Carriot et al., 2014*; *Carriot et al., 2017*). Thus, an important direction for future studies will be to consider the effects of potential motor synergies in the encoding of self-motion by central vestibular neurons.

## Materials and methods

### Surgical procedures and data acquisition

All experimental protocols were approved by the McGill University Animal Care Committee and were in compliance with the guidelines of the Canadian Council on Animal Care. Three male macaque monkeys (2 *Macaca mulatta* and 1 *Macaca fascicularis*) were prepared for chronic extracellular recording using aseptic surgical techniques as previously described (*Massot et al., 2011*). Animals (aged 7, 8, and 8 years old) were housed in pairs on a 12 hr light/dark cycle.

*Head movement recording.* In the first stage of this study, head movements of freely moving animals were recorded as described previously (*Carriot et al., 2017*). Briefly, head movement recordings were made using a microelectromechanical systems (MEMS) module (iNEMO platform, STEVAL-MKI062V2; STMicroelectronics) that was fixed to the animal's head. The MEMS module recorded linear acceleration in all three axes of motion (i.e. fore-aft, lateral, and vertical) as well as angular velocity for rotations along each of these. Each monkey was then released into a large expansive space (240 m$^3$) where it was able to freely move (e.g. forage, groom, walk, run) and interact with another monkey from our colony. Enrichment materials (scattered treats, toys, etc.) as well as multiple structures that the monkeys could climb, were distributed throughout this space (for details see: *Carriot et al., 2017*).

*Single-unit recordings.* In the second stage of this study, we performed electrophysiology experiments in which we recorded the extracellular activity of horizontal semicircular canal afferents and central vestibular-only (VO) neurons. Head-restrained monkeys were seated in a primate chair that was mounted on a motion platform rotating about the vertical axis (i.e. yaw rotation). The motion platform produced rotations such that the head angular velocity stimulus recorded by a gyroscope

mounted on the animal's head matched those experienced during natural activities described above (*Figure 1—figure supplement 1A*, compare blue and red traces). We then applied rotational stimuli whose time course and spectral frequency content matched those of the recorded head movements (*Figure 1B and C*, and S1; see Materials and methods). We found that that the probability distributions of these signals were well-fit by a Gaussian (*Figure 1—figure supplement 1B*). We also recorded the extracellular activity of the same vestibular afferents and central neurons during sinusoidal rotation stimuli delivered at frequencies of 0.5–16 Hz with peak velocities of 40°/s. Data was collected through the Cerebus Neural Signal Processor (Blackrock Microsystems). Action potentials were discriminated from extracellular recordings offline (Offline Sorter, Plexon).

To confirm that each neuron in our sample discharged in a manner consistent with previous analyses, responses were characterized during voluntary eye movements and passive whole-body rotations. Monkeys were trained to track a small visual target (HeNe Laser) projected onto a white cylindrical screen located 60 cm away from the head for a juice reward, and eye position was measured using the magnetic search-coil technique. Both afferents and VO cells in our dataset were unresponsive to saccadic eye movements, smooth pursuit, and ocular fixation made to track the target. We note that, while horizontal canal afferents only respond to the component of rotational self-motion along the yaw axis (*Fernandez and Goldberg, 1971*), this is not the case for central vestibular neurons. Indeed, these neurons integrate inputs from multiple canal and otoliths in a manner that is both nonlinear and frequency dependent (*Dickman and Angelaki, 2002*; *Carriot et al., 2015*). Thus, we only considered yaw rotations here in order to directly compare the response properties of horizontal canal afferents to those of their central neuron targets.

## Analysis of neuronal discharges

Data were imported into MATLAB (MathWorks) for analysis using custom-written algorithms (*Mitchell et al., 2018*). Head velocity signals were sampled at 1 kHz and digitally filtered at 125 Hz. Previous studies have established that vestibular afferents display large heterogeneities in terms of their resting discharge. In particular, the resting discharge regularity of vestibular afferents displays a bimodal distribution, which defines two afferent classes, regular and irregular, further characterized by differences in axon diameter as well as response dynamics (reviewed in: *Goldberg, 2000*; *Eatock et al., 2008*). Accordingly, we quantified the regularity of each afferent's resting discharge by computing the normalized coefficient of variation (CV*) (*Massot et al., 2011*). We found a bimodal distribution of CV* values for our data set (Hartigan's Dip Test; p = 0.002) and thus classified afferents with CV*<0.1 as regular and as irregular otherwise, as done previously (*Goldberg et al., 1984*). Spike trains were digitized at 1 kHz. Autocorrelation functions and power spectra were computed from digitized spike trains using the MATLAB functions *xcorr* and *pwelch*, respectively. Power spectra were normalized to their value at 2 Hz. The correlation time was measured by fitting an exponential to the autocorrelation function. The whitening index was computed as the integral of the spike train or stimulus power spectrum from 0 to 20 Hz divided by the difference between the minimum and maximum power multiplied by the frequency range (i.e. 20 Hz). The stimulus power spectrum was computed from the head velocity signal using the MATLAB function *pwelch*.

*Response Dynamics*: To better visualize the firing rate responses of neurons to sinusoidal stimuli at different frequencies, spike trains were convolved by a Kaiser window whose cut-off frequency was set to double that of the stimulus frequency (*Cherif et al., 2008*). Applying these filters does not affect the gain values computed for our dataset (*Cherif et al., 2008*). We verified that the activity of each central neuron recorded from was responsive to head but not eye motion, as noted above (*Roy and Cullen, 2001*). To compute the tuning to sinusoidal head motion, a least-squares regression analysis was used to describe each unit's response to head rotations:

$$\widehat{fr}(t) = b + Gain\ HHV(t+\theta) \tag{1}$$

where $\widehat{fr}(t)$ is the filtered firing rate, $b$ is a bias term set to the unit's resting discharge rate, $HHV(t)$ is the time varying horizontal head velocity, *Gain* is the response sensitivity, and $\theta$ is the phase shift. For each sinusoidal frequency, the values of *Gain* and $\theta$ were determined by maximizing the variance-accounted-for as done previously (*Cullen et al., 1996*), these values are shown as data points in *Figure 5C,D*.

We next fit a transfer function of the following form to the obtained values of *Gain* and $\theta$:

$$H(f) = \frac{AuT_c(1+uT_1)}{(1+uT_c)(1+uT_2)} \qquad (2)$$

where u = 2 $\pi$ i f, *A* is a constant, $T_c$ and $T_2$ are the long and short time constants of the torsion-pendulum model of canal biomechanics and $T_1$ is proportional to the ratio of acceleration to velocity sensitivity of neuron responses, as done previously (*Fernandez and Goldberg, 1971*; *Hullar et al., 2005*; *Massot et al., 2012*). The estimated gain from the transfer function is then given by $|H(f)|$ while the estimated phase shift is given by $\theta_{est} = tan^{-1}(imag[H(f)]/real[H(f)])$. Here real(...) and imag (...) denote the real and imaginary parts of *H*(f), respectively. We fixed $T_c$ to 5.7 s consistent with the Torsion-pendulum model. For each neuron, values of *A*, $T_1$, and $T_2$ were estimated by minimizing the sum of residuals between the observed gain and phase values for sinusoidal stimulation and those computed from the estimated transfer function for those frequencies. Specifically, the sum of residuals was given by:

$$Res = \sum_f (Gain(f) - |H(f)|)^2 + \frac{1}{100} \sum_f (\theta(f) - \theta_{est}(f))^2 \qquad (4)$$

where *Gain* and $\theta$ are obtained using *equation (1)*. The factor of 1/100 for phase was used to make the residuals for gain and phase approximately equal to one another. The sum of residuals was minimized by using the *fminsearch* function in MATLAB. The estimated gain and phase from the transfer function are shown as solid lines in *Figure 5C,D*.

*Response dynamics for naturalistic stimuli:* The time-dependent firing rate was obtained by convolving the spike train with a Kaiser window whose cut-off frequency was set to 40 Hz, which will not affect the power spectrum within the frequency range 0–20 Hz. We found that the firing rate responses of central vestibular neurons were well-fit by a Gaussian (*Figure 1—figure supplement 1B*). For each neuron, we assumed that the time-dependent firing rate is given by:

$$\widehat{fr}_{predicted}(t) = (h * s)(\text{t}) + fr_0 \qquad (5)$$

where $\widehat{fr}_{predicted}(t)$ is the predicted firing rate, "*" denotes a convolution with a filter *h*(t), $fr_0$ is a constant set to the mean firing rate of the resting discharge, and s(t) is the stimulus. The Fourier transform of *h(t)*, $H_1$(f) (i.e., the transfer function), was estimated using:

$$H_1(f) = \frac{P_{rs}(f)}{P_{ss}(f)} \qquad (6)$$

where $P_{rs}(f)$ is the cross-spectrum between the firing rate and the stimulus, and $P_{ss}(f)$ is the stimulus power spectrum. We also estimated a static nonlinearity for each neuron by plotting the experimentally observed time-dependent firing rate as a function of $\widehat{fr}_{estimated}(t)$ and used the following sigmoidal function to fit the data (*Massot et al., 2012*; *Schneider et al., 2015*):

$$T(x) = \frac{c_1}{1 + e^{-c_2(x-c_3)}} \qquad (7)$$

Adding this static nonlinearity to predict the firing rate response to naturalistic stimuli did not alter our results as shown in *Figure 4—figure supplement 2*. Moreover, replacing the transfer function $H_1$(f) by that obtained in response to sinusoidal stimulation (i.e. H(f) from *eq. 2*) also did not alter our results as shown in *Figure 5C,D*. We hypothesize that this is because central vestibular neurons display large resting discharge rates (~50 sp/s) that are highly variable, which effectively reduces the effects of nonlinearities (*Stemmler, 1996*).

We also calculated the transmitted power spectrum using:

$$P_{signal} = P_{ss} \times |H(f)|^2 \qquad (8)$$

This prediction was accurate for afferents as shown in *Figure 4—figure supplement 3* but not for VO neurons as shown in *Figure 4*.

To quantify the amount of information transmitted by vestibular neurons over the behaviorally relevant frequency range (0–20 Hz), we calculated the mutual information rate using the following equation (*Rieke et al., 1996*):

$$MI = \int_0^{20} df \; log_2[1 + SNR(f)] \tag{9}$$

Here, *SNR(f)* is the signal-to-noise ratio given by:

$$SNR(f) = \frac{P_{signal}(f)}{P_0(f)} \tag{10}$$

where $P_0(f)$ is the power spectrum of the resting discharge activity (i.e., in the absence of stimulation). The mutual information rate was then divided by the neural firing rate during stimulation. The normalized mutual information rate was obtained by dividing the actual mutual information by its maximum value obtained by systematically varying the neuronal tuning curve while keeping the resting discharge power spectrum constant for central vestibular neurons. Error estimates for mutual information quantities were determined from the distribution of values obtained from our dataset and are shown as whisker-box plots throughout.

The response power spectrum to the natural stimulus was predicted using linear response theory (*Risken, 1996*):

$$P_R(f) = P_0(f) + |H(f)|^2 P_{ss}(f) \tag{11}$$

We note that linear response theory is generally applicable provided that the system is not near a critical point (e.g., a bifurcation) and that the stimulus perturbation is 'weak' (*Risken, 1996*). Previous studies have successfully used this theory to explain experimental data across multiple systems (*Chacron et al., 2005*; *Huang et al., 2016*).

To maximize information transmission (i.e., optimally encode) under the constraint of a constant noise variance (*Shannon, 1948*; *Rieke et al., 1996*), the following relationship must be satisfied:

$$P_R(f) = C \tag{12}$$

and *C* is a constant. We note that this assumes that the response, stimulus, and resting activity display Gaussian probability distributions. This is the case for the response and stimulus (Fig. 1 – figure supplement 1B) and we furthermore confirmed that probability distributions of the resting time-varying firing rate were well fit by Gaussian distributions for vestibular afferents (R = 0.99±0.005 and 0.98±0.01 for regular and irregular afferents, respectively), as well as for central vestibular neurons (R = 0.83±0.03). To test whether the experimentally obtained response power spectra were constant as a function of frequency, we simulated 1000 independent Poisson processes with the same number of spikes as the experimental data for each neuron. The power spectra obtained for each Poisson spike train were then used to compute the probability distribution of the power for each frequency. We found that the distributions were all Gaussian (Shapiro-Wilk test, p > 0.05 in all cases) as expected from the central limit theorem, and obtained a 95% confidence interval. These are plotted as gray bands in Figures 1 and 2.

## Quantifying trial-to-trial variability to repeated stimulus presentations

For a subset of VO neurons (N = 8), the naturalistic stimulus was repeated multiple (i.e. 2–5) times, allowing us to quantify trial-to-trial variability. For each trial, the spiking response was convolved with a Kaiser window as described above in order to obtain the time-dependent firing rate. We then computed the residual as the difference between the firing rate response to a given trial and the mean response averaged over all trials. The power spectrum of this residual response was then computed as described above and compared to that of the resting discharge for each neuron. Our results, shown in *Figure 6—figure supplement 1*, show that both power spectra were quite similar. We found no significant differences in the estimated power spectrum of the trial-to-trial variability when using only a subset of trials for neurons for which the stimulus was repeated more than twice.

*Modeling the effects of noise and transmitted power on information transmission:* We used a simple model for which we assume that the total response power $P_{response}(f)$ is given by:

$$P_{response}(f) = P_{noise}(f) + P_{tuning,predicted}(f) \tag{13}$$

where $P_{noise}(f)$ is the noise power spectrum and $P_{tuning,predicted}(f)$ is the transmitted power spectrum. We focused on the 0–20 Hz frequency range, corresponding to that of natural self-motion stimuli as noted above. The mutual information rate was computed from (*Rieke et al., 1996*):

$$MI = \int_0^{20} df \, log_2 \left[ 1 + \frac{P_{tuning,predicted}(f)}{P_{noise}(f)} \right] \tag{14}$$

We then considered the following dependencies for the rest and transmitted power spectra:

$$P_{noise}(f) = P_0 + \frac{2 - \frac{f}{20}}{\int_0^{20} df \, \left( 2 - \frac{f}{20} \right)} \tag{15}$$

$$P_{tuning,predicted}(f) = \frac{1 + sensitivity \frac{f}{20}}{\int_0^{20} df \, \left( 1 + sensitivity \frac{f}{20} \right)} \tag{16}$$

where $P_0$ is the background noise power which is completely determined by the noise intensity and *sensitivity* controls the rate at which the transmitted power increases with frequency, while keeping the area under the curve constant and equal to unity. Both $P_0$ and *sensitivity* were varied systematically and their effects on information computed.

## Statistics

Our sample sizes were similar to those generally employed in the field (*Massot et al., 2011*). Before statistical analysis, normality of distribution was evaluated using a Shapiro-Wilk's test. All data were tested for presence of non-stationarities using an augmented Dickey-Fuller test. We did not find any significant non-stationarities during either resting discharge or driven activities for either of afferents or central vestibular neurons ($p > 0.05$ in all cases). To determine if variances between groups were comparable, an F-test was used. Statistical significance ($p < 0.05$) was determined using parametric analysis with either two-tailed Student's *t*-test or one-way ANOVA. Post-hoc pairwise comparisons were conducted using Tukey's honestly significant difference test (*Figure 3*) or Dunnett's test (*Figure 6*). Throughout, values are expressed as mean ±SEM.

## Acknowledgements

This research study was supported by the Canadian Institutes of Health Research (MJC and KEC), Canada research chair program (MJC), and the National Institutes of Health (KEC).

## Additional information

### Funding

| Funder | Grant reference number | Author |
| --- | --- | --- |
| Canadian Institutes of Health Research | | Maurice J Chacron Kathleen E Cullen |
| Canada Research Chairs | | Maurice J Chacron |
| National Institutes of Health | DC2390 | Kathleen E Cullen |

The funders had no role in study design, data collection and interpretation, or the decision to submit the work for publication.

### Author contributions

Diana E Mitchell, Data curation, Formal analysis, Methodology, Writing—original draft, Writing—review and editing; Annie Kwan, Data curation, Formal analysis; Jerome Carriot, Data curation, Investigation, Methodology; Maurice J Chacron, Conceptualization, Data curation, Formal analysis,

Supervision, Methodology, Writing—original draft, Writing—review and editing; Kathleen E Cullen, Conceptualization, Data curation, Supervision, Funding acquisition, Methodology, Writing—original draft, Writing—review and editing

## Author ORCIDs

Diana E Mitchell  http://orcid.org/0000-0003-0733-484X
Maurice J Chacron  http://orcid.org/0000-0002-3032-452X
Kathleen E Cullen  http://orcid.org/0000-0002-9348-0933

## Ethics

Animal experimentation: All experimental protocols were approved by the McGill University Animal Care Committee (#2001-4096) and were in compliance with the guidelines of the Canadian Council on Animal Care. Three male macaque monkeys (2 *Macaca mulatta* and 1 *Macaca fascicularis*) were prepared for chronic extracellular recording using aseptic surgical techniques as previously described (Massot et al., 2011). Animals (aged 7, 8, and 8 years old) were housed in pairs on a 12 hour light/dark cycle.

## Decision letter and Author response

Decision letter https://doi.org/10.7554/eLife.43019.019
Author response https://doi.org/10.7554/eLife.43019.020

# Additional files

## Supplementary files

• Transparent reporting form
DOI: https://doi.org/10.7554/eLife.43019.015

## Data availability

All data have been deposited on Figshare under the URL https://doi.org/10.6084/m9.figshare.7423724.v1.

The following dataset was generated:

| Author(s) | Year | Dataset title | Dataset URL | Database and Identifier |
|---|---|---|---|---|
| Mitchell D, Kwan A, Carriot J, Chacron M, Cullen KE | 2018 | Figure source data for "Neuronal variability and tuning are balanced to optimize coding of naturalistic self-motion in primate vestibular pathways" | https://doi.org/10.6084/m9.figshare.7423724.v1 | Figshare, 10.6084/m9.figshare.7423724.v1 |

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
