## [Decision Letter]

[Editors’ note: a previous version of this study was rejected after peer review, but the authors submitted for reconsideration. The first decision letter after peer review is shown below.]

Thank you for submitting your work entitled "Neuronal variability and tuning are balanced to optimize coding of naturalistic self-motion in early vestibular pathways" for consideration by *eLife*. Your article has been reviewed by three peer reviewers and the evaluation has been overseen by a Reviewing Editor and a Senior Editor. The reviewers have opted to remain anonymous.

Our decision has been reached after consultation between the reviewers. One central issue emerged in the review process, and was emphasized in the discussions among reviewers: does the analysis in the paper provide an accurate estimate of the noise spectrum? This is a critical issue for the work described since the paper argues that the signal and noise properties together are nicely matched to the spectral properties of natural vestibular inputs. In the work described, noise is estimated from the spontaneous discharge of a cell (i.e. in the absence of any stimulus). The reviewers felt that it was essential to consider whether the noise might depend on the stimulus, or if it is in fact independent of the stimulus as assumed. In particular, any shaping of the noise by the stimulus would substantially alter interpretation of the results. Checking for stimulus-dependent noise likely would require analyzing responses to repeated stimulation with the same stimulus, and, e.g. comparing the fluctuations about the mean response with the spontaneous discharge. This point, along with several other points of varying levels of importance, is highlighted in the individual reviews below.

Based on these discussions and the individual reviews below, we regret to inform you that your work will not be considered further for publication in *eLife* unless this central issue of the potential stimulus dependence of the noise can be addressed. If you are able to test for such stimulus dependence and its impact on your conclusions, we would be happy to reconsider the paper.

*Reviewer #1:*

This paper investigates coding in the vestibular system, particularly whether such coding matches predictions from the efficient coding hypothesis. The main contribution of the paper is to incorporate noise into tests of efficient coding, unlike a good deal of past work which has emphasize tuning properties but neglected noise. The basic results of the paper are clearly illustrated and described. I have a few concerns about how much of an advance the work in the paper represents and about some of the technical aspects of the analysis.

1) Novelty. A number of past studies have considered the impact of noise on efficient coding strategies (e.g. Tkacik et al., 2010, Doi et al., 2012, and Kastner et al., 2015), and several of these incorporate experimental data. Thus I am not sure the statement that this is the first experimental test of the impact of noise on efficient coding (e.g. in Abstract) fairly represents the state of the field. A related issue comes up at the end of the Introduction (also at the end of the Results): I'm not sure it is entirely accurate to say that the prevailing view is that efficient coding depends only on tuning given that many papers that consider noise or thresholding (e.g. Pitkow and Meister, 2012).

2) Noise. An implicit and untested assumption in the paper is that the noise does not depend on the stimulus – and hence that noise can be estimated from the spontaneous discharge. This is quite important to verify as stimulus dependence of the noise could change the results.

*Reviewer #2:*

The paper examines efficient coding in the vestibular system. The authors use naturalistic stimulation (with a roughly power-law distribution in the range [0, 20Hz]) and record from VO neurons to show that the output of VO neurons is spectrally nearly white, consistent with efficient coding predictions. They show that most of the whitening occurs in VO neurons, not in their afferents. Furthermore, by analyzing VO neuron transfer functions they show that the data can be explained only if they take into account the spectrum of noise which is not flat, and which the authors estimate by proxy from the spectrum of spontaneous activity ("resting discharge variability").

Overall, I find the paper well-written and convincing. I have some reservations and suggestions for improvement (below), but overall am supportive of the publication. The hypothesis is clear and while the approach is not novel in the wider area of sensory coding (i.e., early works in the visual and auditory pathways), the test is carried out well and is, to my knowledge [I am not an expert in the vestibular system], the first convincing demonstration in the vestibular system. Additionally, it illustrates the importance of taking into account the spectral properties of the noise. Thus, I think it is of sufficient relevance to support *eLife* publication.

In their argument, the authors equate the spectrum of the resting discharge (= response of VO cells in absence of any stimulation) with the noise power spectrum (= spectrum of fluctuations around the mean response over identical stimulus presentations). There is no direct measurement of the noise power spectrum; for this, the authors would need identical repeats of the naturalistic stimulus, which (so far as I can tell) are not used in the paper. I have serious concerns about replacing true noise power estimates with the power of spontaneous activity.

Neurons could easily exhibit activity that is in the non-stimulus-driven regime substantially different statistically than when driven, invalidating the author's logic. The authors spell out this assumption in the first paragraph of the subsection “Impact of resting discharge variability on coding of self-motion in the vestibular system”, and cite theoretical studies of Charcron et al. as support. While true that stimulus-driven activity needs to perturb resting discharge to be detected, that is mathematically not completely equivalent to equating resting discharge spectrum to noise, and it is simply an empirical question whether these two spectra are actually the same.

As a small indication of the problems that may come from equating the spectrum of resting discharge with the noise spectrum, if I look in detail at Figure 6B, the resting discharge power is actually larger than the total response power at very low frequencies (close to DC), leading authors to overpredict at least the DC power in 6B. Can the authors provide estimates of true noise power or support their claim that resting discharge variability is a good proxy for it? [Detail: power spectra in paper figures say "Normalized power", but I couldn't find what is normalization (I only found references to "normalized noise" and "normalized MI"). The theory of linear filtering / whitening is formulated in terms of absolute spectra, not normalized, and without understanding the normalization, I cannot figure out how to relate the theory to the plotted data.]

Other issues:

1) Introduction. The authors emphasize that previously non-trivial shapes of noise spectra were not taken into account (Introduction, second paragraph etc.). I find this claim a bit overblown, also in Results ("thereby overturning the prevailing view that whitening is achieved by neural tuning alone"). Whitening work in early vision certainly did take into account real noise spectra (not assuming flat), e.g., van Hateren, Ruttiger, Sun, Lee (J Neurosci, 22, 2002), although it did not emphasize this point. Applications of efficient coding to data outside neuroscience specifically established the crucial role for efficient representation of detailed structure of noise (Tkacik, Callan and Bialek, 2008). I do think the authors should emphasize the importance of the frequency structure of the noise, but tone down the tone somewhat. It is a nice paper, without overturning the paradigm.

Regarding other citations: the theoretical framework used by the authors for efficient coding / whitening, also with variable noise power spectrum, has been developed early by JH van Hateren, 1992.

2) The authors restrict their analysis to [0,20] Hz for both inputs and output firing rates, claiming that this is the representative and relevant frequency range. Can you include a distribution of firing rates, maybe as an inset in 1D, in linear or log scale, as appropriate? Also, the distribution of inputs? Also, is it possible to see the full stimulus/response spectra (not restricted to [0,20]), perhaps in the supplementary information, so that the readers can put the [0,20] Hz range in context?

3) Subsection “Response dynamics”. I'm not sure I follow the logic. You say you infer gain and theta, but Equation 2 is an equation for H(f). So it is unclear to me if you first obtained H from Equation 6, and then fitted it using Equation 2, and then extracted gain and theta (which goes in the opposite order of your methods), or did you do it by some other means. Please clarify. Also, as written, gain and theta formulae (following Equation 2) do not make sense, since they take the complex argument of H, which by your definition in Equation 2 is a real number (as it is in Equation 6, since it is the ratio of power spectra).

4) On first reading I found it difficult to understand what the Figures 5B and 5C, and the text that goes with them, are telling me. Upon reading again, I found out that they tell a pretty simple thing but possibly in a complicated way. Perhaps this would be easiest if somehow B and C (and E and F) could be visually combined, to indicate that B and E represent possible variations and C and F the resulting response, and to indicate in these figures what are the "true" cases considered (e.g. b in panels B and C), perhaps indicated by full lines, and what are the synthetic cases (increasing / decreasing slope away from the true slope), maybe shown in dashed lines.

5) Discussion. I find these two formulations highly dubious.a) "spectral frequency content of neural variability significantly contributes to whitening of neural responses" and

b) "We hypothesize that increased resting discharge variability, while detrimental at the single neuron level, is actually beneficial at the population level by lowering neural correlations that affect decoding".

Both imply that it is somehow the noise properties that are optimized to contribute to efficiency, but this feels like turning the logic on its head. Current thinking is that noise is the consequence of fundamental limits/constraints, and the mean responses are something that neurons can adjust. This is simply because if neurons could adjust the noise arbitrarily, they should simply tune it to zero and achieve perfect transmission. Instead, *given* noise, they adjust what they can, which is the tuning function. I would rather emphasize that optimality (= whitening regime of efficient coding) requires a precise match between the input distribution (power law), the tuning function, and the noise spectrum, and that you have successfully demonstrated such matching in the vestibular system.

Claim b) is even more difficult to support and I would remove it from the paper entirely. It is true that adding independent noise to decorrelate will, in the limit of high noise, decorrelate the responses, but also lower the information transmission to zero, so that is a useless way to decorrelate. First, you'd have to establish that the noise actually is independent across neurons, which goes beyond the scope of the paper. Second, "tuning" of the coupling and noise correlations to increase the information coding has long been explored in theory, and if neurons actually have substantial intrinsic noise, the result is not that they should be decorrelated (Barlow, Redundancy reduction revisited; Tkacik et al., 2010). Third, whether correlations hurt or facilitate decoding depends on the exact structure of correlations and stimuli (de Silveira, Berry MJ, PLOS Comput Biol 10, 2014). So either you'd need to discuss your claim more in depth, or remove it from the Discussion.

*Reviewer #3:*

This paper provides some of the first experimental evidence for a particular facet of efficient coding theory, namely the "water-filling analogy" wherein the allocation of response states respects both the distribution of external stimuli and the noise inherent in the neural channel encoding the stimulus. By recording both from vestibular afferents and their central targets, the authors demonstrate that whitening is achieved in central vestibular neurons, but is not simply inherited from their inputs. The noise profile of VO neurons contributes to whitening, which is not achieved by tuning alone. The paper also provides the first clear demonstration of whitening in the vestibular pathway, in response to naturalistic inputs.

Overall, the manuscript is clear, detailed, and describes results that will be of broad interest to the neuroscience community. The figures are clear and well-organized.

A few comments that could improve the impact and/or clarity of the paper:

1) In the Discussion, the part about downstream decoding could use a little more explanation. As it stands, the argument seems to be that while noise reduces single VO information rates, it's sculpted so that it whitens their response profiles and therefore reduces correlations between neurons. Downstream, neurons can more effectively average out the noise if neurons are uncorrelated. How large is this effect? A simple model could demonstrate the magnitude of this decoding benefit and would strengthen this part of the Discussion.

2) Some notes about error measures on the information quantities calculated through the paper would be useful.

3) The stepwise whitening from regular to irregular afferents to VO neurons is intriguing, but no detailed speculation is made about the first step from regular to irregular afferents. What can be said about this transformation and its circuit or synaptic underpinnings?

[Editors’ note: what now follows is the decision letter after the authors submitted for further consideration.]

Thank you for submitting your article "Neuronal variability and tuning are balanced to optimize coding of naturalistic self-motion in early vestibular pathways" for consideration by *eLife*. Your article has been reviewed by three peer reviewers and the evaluation has been overseen by a Reviewing Editor and Joshua Gold as the Senior Editor. The reviewers have opted to remain anonymous.

The reviewers have discussed the reviews with one another and the Reviewing Editor has drafted this decision to help you prepare a revised submission.

As you will see in the individual reviews below, the reviewers all agreed that your revisions improved the paper substantially. Nonetheless, several substantive issues remain. In discussion, all three reviewers agreed that these were important issues that needed to be dealt with before making a final decision on the paper.

*Reviewer #1:*

This paper investigates coding of naturalistic head movements in the vestibular system. In particular, the paper tests predictions of efficient coding theories that efficient coding should minimize temporal correlations in the stimuli. The inclusion of data showing that the noise estimated from spontaneous activity is a good estimate of that during a stimulus strengthens the argument in the paper considerably. I have a few remaining concerns, largely about how the work is presented, and some smaller suggestions for clarity:

Presentation of tuning functions. The tuning curves in Figure 4B need to be connected to the raw data more thoroughly and the assumptions that go into their estimation stated more clearly. It would be helpful to compare the tuning measured with naturalistic stimuli with that measured with sine waves (comparing Figure 4 and Figure 5 they look substantially different).

Use of "optimal coding." The paper used phrases like "coding is optimized" often, especially in regard to the information calculations but in many other places as well. I think these are much too broad and "optimal" needs to be clearly defined. One specific issue here is that, assuming I am not missing something, the main message of the paper is that noise is added between the primary afferents and VO neurons that whitens the responses. Adding this noise certainly does not improve information transmission. Nor does the noise that is added have minimal impact (e.g. noise could be added entirely at frequencies > 100 Hz, and that would have negligible impact on coding). So the situation described is more subtle and I think does not fit under a broad definition of optimal coding. This issue needs to be treated more carefully throughout the paper.

Repeated stimulus trials. The data using the repeated stimulus trials was analyzed to check that the residual responses during these trials provided an estimate of noise consistent with the measured spontaneous activity. But doesn't that data also provide a way to check the entire analysis, free of any modeling assumptions? The mean response across repeated trials should estimate the "signal" power spectrum and the residuals the noise spectrum. This would provide an alternative to the tuning-curve based approach.

*Reviewer #2:*

I'm satisfied with the way the authors' efforts to modify the manuscript, with the bit of the remaining doubt of whether 2-5 stimulus repeats (subsection “Quantifying trial-to-trial variability to repeated stimulus presentations”) are sufficient to really extract reliably the noise power spectrum that was the major sticking point of the first submission.

*Reviewer #3:*

The thorough revisions have satisfied all of my concerns, and the addition of the data for repeated trials in Figure 6—figure supplement 1 greatly strengthens the work.

Given the substantial and substantive revision of the Introduction and Discussion, I think this sentence in the Abstract should also be revised:

"Here, we provide the first experimental evidence supporting this view by recording from neurons in early vestibular pathways during naturalistic self-motion."

Suggestion: either add more specific details to support the use of "first" or simply omit "the first".

[Editors' note: further revisions were requested prior to acceptance, as described below.]

Thank you for submitting your article "Neuronal variability and tuning are balanced to optimize coding of naturalistic self-motion in primate vestibular pathways" for consideration by *eLife*. Your article has been read by Fred Rieke as Reviewing Editor and Joshua Gold as the Senior Editor.

The paper in general is in good shape and the changes made in response to reviewer comments have clarified several issues. One issue remains in a few places regards whether the signaling properties you identify maximize information. The caveats to that statement are described nicely in the Discussion (subsection “Impact of resting discharge variability on coding of self-motion in the vestibular system”). I think a similar more nuanced statement about whether information is maximized (and hence whether coding is optimal) should appear in the Abstract (last two sentences) and Introduction (last paragraph) – where is it stated simply that information is maximized (which would be achieved by not adding any noise rather than by whitening).

---

## [Author Response]

[Editors’ note: the author responses to the first round of peer review follow.]

Reviewer #1:[…] 1) Novelty. A number of past studies have considered the impact of noise on efficient coding strategies (e.g. Tkacik et al., 2010, Doi et al., 2012, and Kastner et al., 2015), and several of these incorporate experimental data. Thus I am not sure the statement that this is the first experimental test of the impact of noise on efficient coding (e.g. in Abstract) fairly represents the state of the field. A related issue comes up at the end of the Introduction (also at the end of the Results): I'm not sure it is entirely accurate to say that the prevailing view is that efficient coding depends only on tuning given that many papers that consider noise or thresholding (e.g. Pitkow and Meister, 2012).

We have rewritten the Introduction and Discussion to better emphasize the difference between our findings and those of previous studies. Specifically, we now mention in the Introduction that, although previous theoretical studies have predicted that the trial-to-trial variability of single neuron responses can make a substantial contribution to optimal coding (Rieke et al., 1996; Tkacik et al., 2008; van Hateren, 1992), experimental studies have either not explicitly investigated the effect of such variability on optimized coding (van Hateren et al., 2002) or have found minimal effects (Pitkow and Meister, 2012). Moreover, we now cite the prior studies (Kastner et al., 2015; Doi et al., 2012) noted above. Specifically, we consider these in the Discussion and emphasize that they instead focused on the role of variability of neural responses across a population (rather than at the single neuron level) towards optimizing information coding.

2) Noise. An implicit and untested assumption in the paper is that the noise does not depend on the stimulus – and hence that noise can be estimated from the spontaneous discharge. This is quite important to verify as stimulus dependence of the noise could change the results.

We appreciate the reviewer’s concern. To directly address this important point, we have performed additional experiments where recordings from VO neurons were obtained and the naturalistic stimulus was repeated multiple times, thereby allowing us to quantify response trial-to-trial variability. We found that the trial-to-trial variability could indeed be estimated from the resting discharge, as both display similar power spectra (Figure 6—figure supplement 1A). Consequently, using the variability power spectrum rather than that of the resting discharge also correctly predicted temporal whitening in response to naturalistic self-motion (Figure 6—figure supplement 1B, C). This is now stated in the Results (subsection “Whitening of natural self-motion stimuli by central vestibular neurons can be explained by taking into account both their tuning properties and their resting discharge variability”, last paragraph) and Materials and methods (subsection “Quantifying trial-to-trial variability to repeated stimulus presentations”).

Reviewer #2:[…] In their argument, the authors equate the spectrum of the resting discharge (= response of VO cells in absence of any stimulation) with the noise power spectrum (= spectrum of fluctuations around the mean response over identical stimulus presentations). There is no direct measurement of the noise power spectrum; for this, the authors would need identical repeats of the naturalistic stimulus, which (so far as I can tell) are not used in the paper. I have serious concerns about replacing true noise power estimates with the power of spontaneous activity.Neurons could easily exhibit activity that is in the non-stimulus-driven regime substantially different statistically than when driven, invalidating the author's logic. The authors spell out this assumption in the first paragraph of the subsection “Impact of resting discharge variability on coding of self-motion in the vestibular system”, and cite theoretical studies of Charcron et al. as support. While true that stimulus-driven activity needs to perturb resting discharge to be detected, that is mathematically not completely equivalent to equating resting discharge spectrum to noise, and it is simply an empirical question whether these two spectra are actually the same.As a small indication of the problems that may come from equating the spectrum of resting discharge with the noise spectrum, if I look in detail at Figure 6B, the resting discharge power is actually larger than the total response power at very low frequencies (close to DC), leading authors to overpredict at least the DC power in 6B. Can the authors provide estimates of true noise power or support their claim that resting discharge variability is a good proxy for it?

We appreciate the reviewer’s concern. As noted above in response to reviewer 1, we have performed additional experiments where recordings from VO neurons were obtained and the naturalistic stimulus was repeated multiple times, thereby allowing us to quantify response trial-to-trial variability. We found that the trial-to-trial variability could indeed be estimated from the resting discharge, as both displayed similar power spectra (Figure 6—figure supplement 1A). Consequently, using the variability power spectrum rather than that of the resting discharge also correctly predicted temporal whitening in response to naturalistic self-motion (Figures S6B, S6C). This is now stated in the Results (subsection “Whitening of natural self-motion stimuli by central vestibular neurons can be explained by taking into account both their tuning properties and their resting discharge variability”, last paragraph) and Materials and methods (subsection “Quantifying trial-to-trial variability to repeated stimulus presentations”).

[Detail: power spectra in paper figures say "Normalized power", but I couldn't find what is normalization (I only found references to "normalized noise" and "normalized MI"). The theory of linear filtering / whitening is formulated in terms of absolute spectra, not normalized, and without understanding the normalization, I cannot figure out how to relate the theory to the plotted data.]

We now explain the rationale for normalization in the manuscript. Specifically, all power spectra were normalized to their value at 2 Hz and we now explain this in the Materials and methods section of the revised text (subsection “Analysis of Neuronal Discharges”). For simplicity, we elected to show normalized power spectra in order to be able to better compare those of neural responses and the stimulus (e.g., Figures 1E, 1F). Regarding both experiments and theory, we note that normalization does not affect values obtained for the measures of temporal whitening (i.e., white index and correlation time).

Other issues:1) Introduction. The authors emphasize that previously non-trivial shapes of noise spectra were not taken into account (Introduction, second paragraph etc.). I find this claim a bit overblown, also in Results ("thereby overturning the prevailing view that whitening is achieved by neural tuning alone"). Whitening work in early vision certainly did take into account real noise spectra (not assuming flat), e.g., van Hateren, Ruttiger, Sun, Lee (J Neurosci, 22, 2002), although it did not emphasize this point. Applications of efficient coding to data outside neuroscience specifically established the crucial role for efficient representation of detailed structure of noise (Tkacik, Callan and Bialek, 2008). I do think the authors should emphasize the importance of the frequency structure of the noise, but tone down the tone somewhat. It is a nice paper, without overturning the paradigm.Regarding other citations: the theoretical framework used by the authors for efficient coding / whitening, also with variable noise power spectrum, has been developed early by JH van Hateren, 1992.

We have rewritten the Introduction to better emphasize the difference between our findings and those of previous studies. Specifically, we now mention in the Introduction that, although previous theoretical studies have predicted that the trial-to-trial variability of single neuron responses can make a substantial contribution to optimal coding (Rieke et al. 1996; Tkacik et al., 2008; van Hateren, 1992), experimental studies have either not explicitly investigated the effect of such variability on optimized coding (van Hateren et al., 2002) or have found minimal effects (Pitkow and Meister, 2012).

2) The authors restrict their analysis to [0,20] Hz for both inputs and output firing rates, claiming that this is the representative and relevant frequency range. Can you include a distribution of firing rates, maybe as an inset in 1D, in linear or log scale, as appropriate? Also, the distribution of inputs? Also, is it possible to see the full stimulus/response spectra (not restricted to [0,20]), perhaps in the supplementary information, so that the readers can put the [0,20] Hz range in context?

We have added the stimulus spectra showing power at frequencies ranging up to 50 Hz (inset in Figure 1—figure supplement 1A), confirming that the spectral power of natural self-motion is negligible at frequencies >20 Hz. We have also included a distribution of firing rate and head velocity stimulus in Figure 1—figure supplement 1B as requested (subsection “Central vestibular neurons display temporal whitening in response to naturalistic self-motion”, first paragraph).

3) Subsection “Response dynamics”. I'm not sure I follow the logic. You say you infer gain and theta, but Equation 2 is an equation for H(f). So it is unclear to me if you first obtained H from Equation 6, and then fitted it using Equation 2, and then extracted gain and theta (which goes in the opposite order of your methods), or did you do it by some other means. Please clarify. Also, as written, gain and theta formulae (following Equation 2) do not make sense, since they take the complex argument of H, which by your definition in Equation 2 is a real number (as it is in Equation 6, since it is the ratio of power spectra).

We thank the reviewer for bringing this to our attention and have rewritten the Materials and methods section to clarify the logic of our approach. Specifically, we now mention that we first obtained gain and phase values for different sinusoidal stimulation frequencies using Equation 1. Then, we fit a transfer function of the form given by Equation 2 to the data and then estimated the gain and phase from this function. The gain was then used to predict responses to naturalistic self-motion.

4) On first reading I found it difficult to understand what the Figures 5B and 5C, and the text that goes with them, are telling me. Upon reading again, I found out that they tell a pretty simple thing but possibly in a complicated way. Perhaps this would be easiest if somehow B and C (and E and F) could be visually combined, to indicate that B and E represent possible variations and C and F the resulting response, and to indicate in these figures what are the "true" cases considered (e.g. b in panels B and C), perhaps indicated by full lines, and what are the synthetic cases (increasing / decreasing slope away from the true slope), maybe shown in dashed lines.

We thank the reviewer for these suggestions and have modified the figure accordingly. Please note that, as per reviewer 1, we have also switched Figure 5 and Figure 4—figure supplement 4. In addition, we have reworked the text pertaining to this Figure in the results.

5) Discussion. I find these two formulations highly dubious.a) "spectral frequency content of neural variability significantly contributes to whitening of neural responses" andb) "We hypothesize that increased resting discharge variability, while detrimental at the single neuron level, is actually beneficial at the population level by lowering neural correlations that affect decoding".Both imply that it is somehow the noise properties that are optimized to contribute to efficiency, but this feels like turning the logic on its head. Current thinking is that noise is the consequence of fundamental limits/constraints, and the mean responses are something that neurons can adjust. This is simply because if neurons could adjust the noise arbitrarily, they should simply tune it to zero and achieve perfect transmission. Instead, given noise, they adjust what they can, which is the tuning function. I would rather emphasize that optimality (= whitening regime of efficient coding) requires a precise match between the input distribution (power law), the tuning function, and the noise spectrum, and that you have successfully demonstrated such matching in the vestibular system.

We understand the reviewer’s comment and have revised this section of the text accordingly. Specifically, we now emphasize throughout that optimal coding requires a precise match between the input distribution (power law), the tuning function, and the noise spectrum.

Claim b) is even more difficult to support and I would remove it from the paper entirely. It is true that adding independent noise to decorrelate will, in the limit of high noise, decorrelate the responses, but also lower the information transmission to zero, so that is a useless way to decorrelate. First, you'd have to establish that the noise actually is independent across neurons, which goes beyond the scope of the paper. Second, "tuning" of the coupling and noise correlations to increase the information coding has long been explored in theory, and if neurons actually have substantial intrinsic noise, the result is not that they should be decorrelated (Barlow, Redundancy reduction revisited; Tkacik et al., 2010). Third, whether correlations hurt or facilitate decoding depends on the exact structure of correlations and stimuli (de Silveira, Berry MJ, PLOS Comput Biol 10, 2014). So either you'd need to discuss your claim more in depth, or remove it from the Discussion.

We agree with the reviewer that our argument was highly speculative and have now revised the text. Specifically, we now hypothesize that a match between stimulus statistics as well as tuning properties and variability will optimize coding of naturalistic self-motion by vestibular neural populations in the context of existing literature.

Reviewer #3:[…] 1) In the Discussion, the part about downstream decoding could use a little more explanation. As it stands, the argument seems to be that while noise reduces single VO information rates, it's sculpted so that it whitens their response profiles and therefore reduces correlations between neurons. Downstream, neurons can more effectively average out the noise if neurons are uncorrelated. How large is this effect? A simple model could demonstrate the magnitude of this decoding benefit and would strengthen this part of the Discussion.

We understand the reviewer’s comment and note that previous theoretical studies have quantified this effect using models (e.g., Figure 3A of Zohary et al. 1994 Nature and Figure 2A of Averbeck et al. 2006 Nat Reviews Neurosci). In light of this comment as well as suggestions by reviewer 2, we have elected to remove this statement from the text.

2) Some notes about error measures on the information quantities calculated through the paper would be useful.

We now state in the Materials and methods that error estimates for mutual information quantities were determined from the distribution of values obtained from our dataset (subsection “Response dynamics for naturalistic stimuli”).

3) The stepwise whitening from regular to irregular afferents to VO neurons is intriguing, but no detailed speculation is made about the first step from regular to irregular afferents. What can be said about this transformation and its circuit or synaptic underpinnings?

We have revised the text to state that regular and irregular afferents comprise parallel channels. Both types of afferents transmit information from the vestibular end organs to the vestibular nuclei. Specifically, this is now mentioned at the beginning of the Results section in relation to Figure 2. Moreover, our results show that the greater whitening seen in irregular afferents originates from their more high-pass tuning curves (Figure 4—figure supplement 3) and this has been further emphasized in the text.

[Editors' note: the author responses to the re-review follow.]

Reviewer #1:This paper investigates coding of naturalistic head movements in the vestibular system. In particular, the paper tests predictions of efficient coding theories that efficient coding should minimize temporal correlations in the stimuli. The inclusion of data showing that the noise estimated from spontaneous activity is a good estimate of that during a stimulus strengthens the argument in the paper considerably. I have a few remaining concerns, largely about how the work is presented, and some smaller suggestions for clarity:Presentation of tuning functions. The tuning curves in Figure 4B need to be connected to the raw data more thoroughly and the assumptions that go into their estimation stated more clearly. It would be helpful to compare the tuning measured with naturalistic stimuli with that measured with sine waves (comparing Figure 4 and Figure 5 they look substantially different).

We thank the reviewer for this comment and note that we used tuning curves obtained under both naturalistic and sinusoidal stimulation in order to account for any possible response nonlinearities during naturalistic stimulation. Our results show that using either tuning curves cannot account the experimentally observed whitening (Figures 4, 5). To address the reviewers’ concern, we have now included an inset in Figure 5D comparing the tuning curves measured with naturalistic stimuli and those measured with sine waves. We note that, at lower frequencies, the tuning measured from naturalistic stimulation is lower than that measured using sinewaves (Figure 5D, compare gray and green curves). This is expected because central vestibular neurons display a boosting nonlinearity characterized by attenuation of the sensitivity to low frequencies in the presence of high frequencies (Massot et al., 2012). We now mention this in the legend of Figure 5.

Use of "optimal coding." The paper used phrases like "coding is optimized" often, especially in regard to the information calculations but in many other places as well. I think these are much too broad and "optimal" needs to be clearly defined. One specific issue here is that, assuming I am not missing something, the main message of the paper is that noise is added between the primary afferents and VO neurons that whitens the responses. Adding this noise certainly does not improve information transmission. Nor does the noise that is added have minimal impact (e.g. noise could be added entirely at frequencies > 100 Hz, and that would have negligible impact on coding). So the situation described is more subtle and I think does not fit under a broad definition of optimal coding. This issue needs to be treated more carefully throughout the paper.

We have revised the paper to explicitly refer to optimal coding in the context that the response power spectrum is independent of frequency (Figures 1E, F) and that the mutual information is close (>90%) to its maximum value (Figure 6D). We have clarified that the addition of noise does not increase information but that the noise structure will influence optimal coding. We have further clarified the Discussion to make it clear that, although the noise in central vestibular neurons does not increase mutual information (see Figure 4—figure supplement 3G and Figure 6—figure supplement 2), the structure of this noise is such that the mutual information is close (>90%) to its maximum value (Figure 6D).

Repeated stimulus trials. The data using the repeated stimulus trials was analyzed to check that the residual responses during these trials provided an estimate of noise consistent with the measured spontaneous activity. But doesn't that data also provide a way to check the entire analysis, free of any modeling assumptions? The mean response across repeated trials should estimate the "signal" power spectrum and the residuals the noise spectrum. This would provide an alternative to the tuning-curve based approach.

We have analyzed our data in the proposed way and have added the results to Figure 6—figure supplement 1. Our results show that the response spectrum can be well predicted using the mean response and the residual.

Reviewer #2:I'm satisfied with the way the authors' efforts to modify the manuscript, with the bit of the remaining doubt of whether 2-5 stimulus repeats (subsection “Quantifying trial-to-trial variability to repeated stimulus presentations”) are sufficient to really extract reliably the noise power spectrum that was the major sticking point of the first submission.

We understand the reviewer’s concern. To address this point, we compared estimates of the variability power spectrum obtained using a subset of stimulus repetitions (i.e., 2 vs. 3 vs. 4 vs. 5) to those obtained using all stimulus repetitions and found no significant differences for our dataset. This is now stated in the Materials and methods (subsection “Quantifying trial-to-trial variability to repeated stimulus presentations”).

Reviewer #3:The thorough revisions have satisfied all of my concerns, and the addition of the data for repeated trials in Figure 6—figure supplement 1 greatly strengthens the work.Given the substantial and substantive revision of the Introduction and Discussion, I think this sentence in the Abstract should also be revised:"Here, we provide the first experimental evidence supporting this view by recording from neurons in early vestibular pathways during naturalistic self-motion."Suggestion: either add more specific details to support the use of "first" or simply omit "the first".

We thank the reviewer for this suggestion and have updated this sentence accordingly.

[Editors' note: further revisions were requested prior to acceptance, as described below.]

The paper in general is in good shape and the changes made in response to reviewer comments have clarified several issues. One issue remains in a few places regards whether the signaling properties you identify maximize information. The caveats to that statement are described nicely in the Discussion (subsection “Impact of resting discharge variability on coding of self-motion in the vestibular system”). I think a similar more nuanced statement about whether information is maximized (and hence whether coding is optimal) should appear in the Abstract (last two sentences) and Introduction (last paragraph) – where is it stated simply that information is maximized (which would be achieved by not adding any noise rather than by whitening).

We thank the editors for this comment. In the revised manuscript, we make it clear that, while increasing the level of variability will decrease information transmission, changes in the frequency spectrum of variability for a given level can strongly determine optimality of coding (i.e., how close is the mutual information to its maximum value) in the abstract and introduction as requested. In addition, we have changed “maximize” by “optimize” in the Abstract and elsewhere in the Introduction. We felt that this was the best solution given the 150 word limit of the Abstract.